# On Memory: A Comparison of Memory Mechanisms in World Models

## Abstract

World models enable agents to plan within imagined environments by predicting future states conditioned on past observations and actions. However, their ability to plan over long horizons is limited by the effective memory span of the backbone architecture. This limitation leads to perceptual drift in long rollouts, degrading the model's capacity to recall recently observed scenes. In this work, we investigate the effective memory span of transformer-based world models through an analysis of memory augmentation mechanisms. We introduce a taxonomy that distinguishes between memory *encoding* and memory *injection* mechanisms, motivating their roles in extending the world model's memory through the lens of residual stream dynamics. We evaluate twenty combinations of four encoding methods and five injection methods in the MemoryMaze environment. Using a state recall evaluation task across multiple imagination horizons, we measure the memory recall capacity of each mechanism and analyze their respective trade-offs in reconstruction quality, latent prediction error, and computational cost. We further ablate the effect of injection depth and compare the best memory-augmented vision transformer against a pure state-space model backbone. Our central finding is that the mLSTM memory encoder outperforms all alternatives in both reconstruction and latent fidelity metrics. Paired with additive injection, it exhibits the strongest recall capabilities at a moderate computational cost while matching or slightly exceeding a pure Mamba backbone.

## 1 Introduction

World models learn to approximate the dynamics of an environment by predicting future states from the current state and the actions that transform it (Ha and Schmidhuber, 2018; Hafner et al., 2020; 2021; LeCun, 2022; Hafner et al., 2025; Bar et al., 2024; Zhou et al., 2025). By planning within the world model's imagination, an agent can efficiently and safely explore a variety of action trajectories without having to execute them in the slow and failure-prone real environment. This paradigm has proven effective in diverse domains, from Atari games (Hafner et al., 2021; Alonso et al., 2024) to continuous control (Hafner et al., 2020; 2025) and visual navigation (Bar et al., 2024; Zhou et al., 2025).

However, planning within a world model's imagination is currently constrained by short prediction horizons. As the model predicts further into the future, perceptual drift accumulates, resulting in hallucinated or forgotten details. This drift limits the length of useful plans and prevents the model from recalling historical context falling outside the model's context window. Recent work has shown that increasing the memory capacity of the world model can increase the effective prediction horizon of world models (Deng et al., 2023; Samsami et al., 2024; Xiao et al., 2025; Lee et al., 2025; Po et al., 2025). The fundamental goal of these works is to design a mechanism to compress historical context into relevant summaries that are injected into the world model during imagination. One direction is to selectively prepend historical context to the world model's context window using learned or retrieval-based lookup mechanisms (Behrouz et al., 2024; 2025; Xiao et al., 2025). Another approach is to learn a compressed hidden state that summarized relevant historical context, as is done recurrent neural networks Hochreiter and Schmidhuber (1997); Deng et al. (2023); Samsami et al. (2024); Savov et al. (2025); Po et al. (2025).

Each of these approaches explores a different mechanism of modifying the historical information living on the world model's residual stream. A world model's residual stream serves as a multi-channel information highway where compressed knowledge is accumulated through repeated read and write operations across layers (Srivastava et al., 2015; Hochreiter and Schmidhuber, 1997; He et al., 2015; Vaswani et al., 2017; Elhage et al., 2021). Over a long sequence, past information on this stream gets overwritten or diffused as new information is written. Viewing memory mechanisms through the lens of encoding historical context and injecting back into the residual stream can help guide the development of long-horizon world models.

In this study, we explore several memory mechanisms through the lens of how they extract and inject information into the world model's residual stream. Specifically:

1. We introduce a taxonomy that describes memory mechanisms as *encoding* or *injection* methods.

2. We evaluate 20 memory mechanisms on a recent state recall task across multiple imagination horizons.

3. We compare the a memory-augmented transformer against a pure state-space model backbone, testing how hybrid architectures compete against long-context state-space models.

Our central findings are that the mLSTM encoder is the strongest across reconstruction and latent-fidelity metrics over increasingly longer horizons. We also find that the Prepend and Additive methods are the most effective at injecting memory back into the transformer's residual stream, with Additive offering the best quality-computation tradeoff and Prepend the highest short-horizon fidelity.

## 2 Related Work

### 2.1 World Models

World models aim to learn a compact latent representation of environment dynamics for imagination-based control, planning, and prediction. The Dreamer series of work demonstrated that latent imagination and predictive consistency can yield strong policy learning in continuous and discrete domains (Hafner et al., 2020; 2021; 2025). Subsequent works have extended world models to visual reasoning and open-ended settings, such as Diffusion World Models (DIAMOND) (Alonso et al., 2024), Navigation World Models (NWM) (Bar et al., 2024), and DINO-WM leveraging pre-trained vision encoders for zero-shot planning (Zhou et al., 2025). These advances establish world models as a general framework for simulation and planning, but remain constrained by limited context horizons and imperfect long-term consistency.

### 2.2 Memory and Context in Sequence Models

Transformers, while powerful at capturing global dependencies, suffer from quadratic complexity in sequence length. Methods such as Transformer-XL (Dai et al., 2019) introduced segment-level recurrence to extend context windows, while recent work shows that transformers benefit from explicit memory tokens or registers for persistent state tracking (Darcet et al., 2024). Learned test-time memory mechanisms, as explored in Titans and Atlas (Behrouz et al., 2024; 2025), adaptively update external memory buffers, enabling partial alleviation of context limits. However, these approaches primarily target token-level recall rather than the continuous latent dynamics common in visual world modeling.

### 2.3 Memory in Continual Learning

Our encoding and injection decomposition parallels a long-standing distinction in continual learning between how past experience is encoded into a compact memory and how it is later replayed to influence current decisions. Generative-replay methods encode history into the weights of a generative model and recall it by sampling (van de Ven et al., 2020), analogous to our parametric encoders (Titans, and to a lesser extent the recurrent-state encoders), while our injection component plays the role of the replay-recall pathway. Lifelong

place-recognition systems similarly couple a learned memory representation with a recall mechanism for long-horizon spatial consistency (Yin et al., 2023), and adaptive-replay methods study which stored samples to revisit (Smith et al., 2024). Viewing world-model memory through this lens clarifies that encoding and injection mechanisms are separable choices, and that advances in one subfield may transfer to the other.

### 2.4 Memory-Augmented World Models

Explicit memory mechanisms have recently been explored in world models to improve temporal consistency, context lookup capacity, and long-horizon prediction. WorldMem (Xiao et al., 2025) demonstrated that augmenting a transformer-based world model with pose-based retrieval memory mechanism improves generative performance for long-horizon prediction tasks in Minecraft. R2I (Samsami et al., 2024) improved Dreamerv3 by replacing the recurrent state-space model with a modified S4 (Gu et al., 2021) model for long-horizon imagination in complex memory environments. (Deng et al., 2023) compared recurrent, transformer, and state-space backbones, showing that state-space models yield superior long-term retention and stability, and introduced S4WM as a dedicated world model backbone.

### 2.5 State-Space Models for Long-Context Learning

State-space models (SSMs) provide a linear-time alternative to attention, maintaining a hidden state that evolves over time. This approach allows for implicit long-term memory and efficient processing of long sequences. Recent works have integrated SSMs into world modeling pipelines as well. StateSpaceDiffuser (Savov et al., 2025) conditions a diffusion model on the most recent Mamba outputs (Gu and Dao, 2024) using cross-attention, EDELINE (Lee et al., 2025) follows adaptive layer norm Dumoulin et al. (2017); Ho et al. (2022) to steer a diffusion model, and Long-Context State-Space Video World Models (Po et al., 2025) introduce a block-wise SSM scan for improved generative performance. These models demonstrate that SSMs can outperform transformer backbones on extended rollouts and memory benchmarks, yet they often rely on diffusion-based decoders to produce high-fidelity observations. In this work, we analyze whether hybrid vision transformers augmented with various memory mechanisms can capture the benefits of both architectures.

## 3 Residual Stream Framework

### 3.1 The Residual Stream as a Temporal Information Highway

We view memory mechanisms as augmentations to the temporal residual stream $\mathbf{Z}$, the channel through which information flows across both layers and timesteps in a world model. At each point in time, the residual stream encodes short-term contextual information from a finite history $\mathbf{Z}_{t-W:t}$ within a sliding window of size $W$, together with the current spatial context $\mathbf{Z}_t^{(\text{spatial})}$:

$$\mathbf{Z}_t \leftarrow \mathbf{Z}_{t-W:t-1}^{(\text{window})} + \mathbf{Z}_t^{(\text{spatial})}. \tag{1}$$

Here, the spatial features are obtained from an encoder $\text{Enc}(\mathbf{X}_t) \rightarrow \mathbf{Z}_t$, and temporal evolution is modeled by a dynamics function $f(\mathbf{Z}_t, \mathbf{A}_t) \rightarrow \mathbf{Z}_{t+1}$. As the dynamics model processes inputs, each layer writes information to distinct subspaces of the residual stream (Elhage et al., 2021). With a finite window size $W$, the model maintains an array of subspaces that collectively hold information for the most recent $W$ timesteps. Over time, these subspaces are progressively overwritten as new information enters the window.

### 3.2 Memory as Residual Stream Augmentation

Introducing an auxiliary memory mechanism $\mathcal{M}$ can be interpreted as extending or refreshing these subspaces with information from earlier timesteps. Depending on how $\mathcal{M}$ is integrated, it may either restore decayed representations within existing subspaces or create dedicated subspaces specialized for retaining long-term

information, effectively forming a shortcut through time:

$$\mathbf{Z}'_t \leftarrow \mathbf{Z}^{(\text{window})}_{t-W:t-1} + \mathbf{Z}^{(\text{spatial})}_t + g\left(\underbrace{\mathcal{M}(\mathbf{Z}_t)}_{\text{long-past}}\right), \tag{2}$$

where $g(\cdot)$ denotes the transformation or conditioning operation used to integrate the memory output into the residual stream and $\mathcal{M}(\mathbf{Z}_t)$ is conditioned on the current context $\mathbf{Z}_t$. This formulation naturally leads to a decomposition of the memory design space into two orthogonal axes: how past information is *encoded* into the memory, and how that memory is *injected* back into the residual stream.

### 3.3 Decomposing Memory: Encoding and Injection

We categorize the design of memory mechanisms $\mathcal{M}$ into two components: memory encoding and memory injection. The encoding process $\mathcal{M}_{\text{enc}}(\mathbf{Z}_{1:t-1}) \rightarrow \tilde{\mathbf{M}}$ determines how past information is compressed and stored into a compact memory representation $\tilde{\mathbf{M}}$. The injection process defines how $\tilde{\mathbf{M}}$ is integrated back into the model's residual stream, modifying the information available to downstream computations. Each injection method corresponds to a distinct computational mechanism within the transformer block, providing a different "knob" for controlling how historical information influences the model's predictions. A third component, memory selection and retrieval, governs what enters the first two memory mechanisms and is orthogonal to encoding and injection. The selection policy decides which past observations are stored or retrieved. Selection is central to memory in continual learning (van de Ven et al., 2020; Smith et al., 2024) and underlies retrieval-based world models such as WorldMem (Xiao et al., 2025), which selects memory entries by pose similarity. We hold selection fixed, i.e. every encoder observes the full window, in order to isolate the encoding and injection components, and treat learned selection as beyond the present scope. In the following section, we describe each encoding and injection method in detail.

## 4 Memory Mechanisms

### 4.1 Memory Encoders

We explore four approaches to encoding past representations into a compact memory.

**Cache.** The Cache encoder maintains an explicit fixed-size buffer of the $K$ most recently computed latent representations. At each timestep $t$, the current latent $\mathbf{Z}_t \in \mathbb{R}^{N \times d}$ is appended to the buffer and the oldest entry is evicted once the buffer reaches capacity $K$, giving a sliding window of uncompressed past states (Dai et al., 2019; Berges et al., 2024; Xiao et al., 2025; Pouransari et al., 2025):

$$\tilde{\mathbf{M}}_{\text{cache}} = \{\mathbf{Z}_{t-K}, \ldots, \mathbf{Z}_{t-1}\} \in \mathbb{R}^{K \times N \times d}. \tag{3}$$

This provides direct, uncompressed access to recent history at the cost of memory that grows linearly with $K$. Paired with Prepend injection method (Section 4.2), this encoder matches the standard KV-cache, sliding-window mechanism of Transformer-XL (Dai et al., 2019) and WorldMem (Xiao et al., 2025).

**Neural Weights (Titans).** The Titans encoder stores memory in the parameters of a small MLP $f_{\mathbf{W}_t}$, which is updated online at each timestep using a gradient-based surprise signal (Behrouz et al., 2024; 2025). The surprise at timestep $t$ is measured as the prediction error of the current memory state on the new input, and the forget gate $\theta_t$ modulates how much of the previous memory is retained:

$$\mathbf{W}_t = \mathbf{W}_{t-1} - \theta_t \odot \nabla_{\mathbf{W}}\mathcal{L}(f_{\mathbf{W}_{t-1}}, \mathbf{Z}_t), \tag{4}$$

where $\theta_t \in \mathbb{R}$ is the learned forget gate and $\mathcal{L}$ is the surprise loss. Memory is retrieved by a forward pass through the updated network:

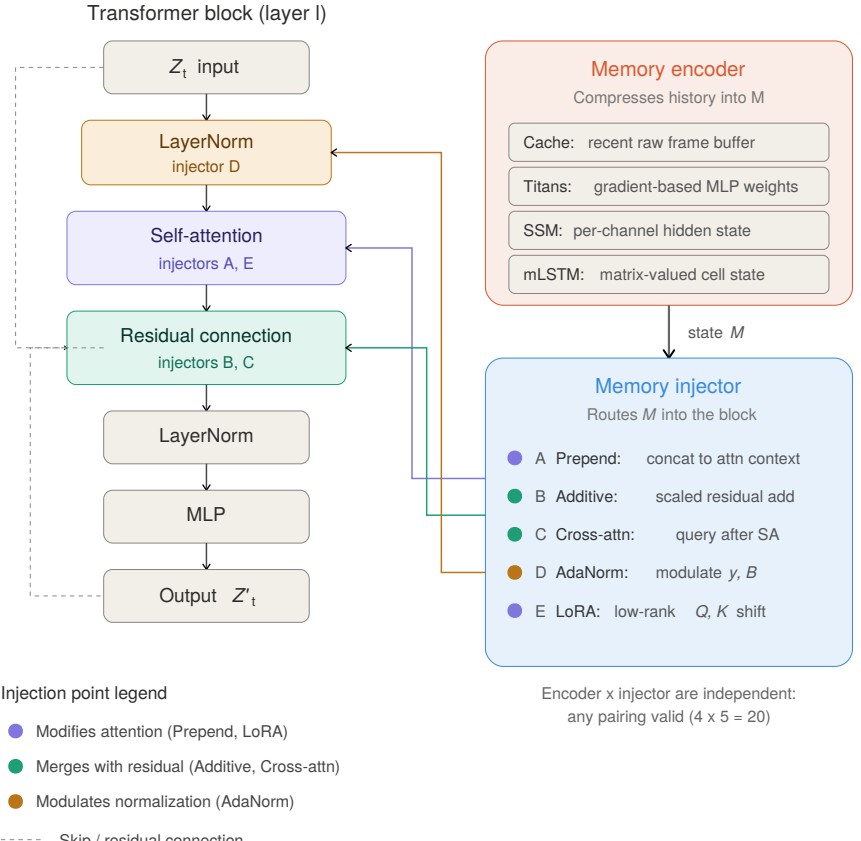

Figure 1: **The modular memory-augmented Transformer framework.** The architecture decouples memory into two independent stages: *encoding* (top right) and *injection* (bottom right). The Memory Encoder compresses historical latent states $Z_{1:t-1}$ into a compact representation $\tilde{M}$ using one of four mechanisms (**Cache**, **Titans**, **SSM**, or **mLSTM**). The Memory Injector then routes $\tilde{M}$ back into the Transformer's residual stream at one of three distinct intervention points, indicated by color: **purple** marks methods that modify the self-attention computation (**Prepend**, **LoRA**); **teal** marks methods that merge directly with the residual connection (**Additive**, **Cross-Attention**); and **amber** marks normalization-based conditioning (**AdaNorm**). Because the two axes are orthogonal, any encoder can be paired with any injector, yielding $4 \times 5 = 20$ combinations. Dashed lines denote skip connections.

$$\tilde{\mathbf{M}}_{\text{titans}} = f_{\mathbf{W}_t}(\mathbf{Z}_t). \tag{5}$$

This enables adaptive, content-dependent compression but requires an inner optimization step at every timestep.

**Recurrent Hidden State (SSM).** The SSM encoder uses an auxiliary Mamba model (Gu and Dao, 2024) to maintain a compressed hidden state over the observation history. At each timestep, the hidden state is updated via the discretized state-space transition with input-dependent parameters $\mathbf{A}_t, \mathbf{B}_t, \mathbf{C}_t$ derived from $\mathbf{Z}_t$:

$$\mathbf{h}_t = \bar{\mathbf{A}}_t \mathbf{h}_{t-1} + \bar{\mathbf{B}}_t \mathbf{Z}_t, \quad \tilde{\mathbf{M}}_{\text{ssm}} = \mathbf{C}_t \mathbf{h}_t, \tag{6}$$

where $\bar{\mathbf{A}}_t$ and $\bar{\mathbf{B}}_t$ are the discretized transition and input matrices respectively, and selectivity is achieved by conditioning all parameters on the input. Crucially, the state transition is applied independently per input channel, with no cross-channel interaction within the hidden state itself.

**Matrix LSTM (mLSTM).** Unlike the scalar hidden state of a standard LSTM (Hochreiter and Schmidhuber, 1997), the mLSTM (Beck et al., 2024) maintains a matrix-valued memory cell $\mathbf{C}_t \in \mathbb{R}^{d \times d}$ that stores key-value associations via an outer product update rule. At each timestep, key and value vectors $\mathbf{k}_t, \mathbf{v}_t \in \mathbb{R}^d$ are derived from $\mathbf{Z}_t$ via learned projections, and the cell is updated as:

$$\mathbf{C}_t = f_t \odot \mathbf{C}_{t-1} + i_t \odot \mathbf{v}_t \mathbf{k}_t^\top, \tag{7}$$

where $f_t = \exp(\tilde{f}_t) \in \mathbb{R}$ and $i_t = \exp(\tilde{i}_t) \in \mathbb{R}$ are exponential forget and input gates derived from learned projections of $\mathbf{Z}_t$. Memory is retrieved via a matrix-vector product with a query vector $\mathbf{q}_t \in \mathbb{R}^d$:

$$\tilde{\mathbf{M}}_{\text{mlstm}} = \frac{\mathbf{C}_t \mathbf{q}_t}{\max(|\mathbf{n}_t^\top \mathbf{q}_t|, 1)}, \quad \mathbf{n}_t = f_t \odot \mathbf{n}_{t-1} + i_t \odot \mathbf{k}_t, \tag{8}$$

where $\mathbf{n}_t$ is a normalizer vector that tracks the cumulative gate activity to stabilize retrieval. Unlike the SSM encoder, the matrix-vector retrieval produces a weighted combination across all memory dimensions, enabling cross-channel interaction during the memory read operation.

## 4.2 Memory Injectors

Once a memory representation $\tilde{\mathbf{M}}$ is obtained, it must be injected into the residual stream. We evaluate five injection strategies, each modifying a different component of the transformer block.

**Context Prepend.** The memory representations are concatenated to the beginning of the input token sequence before self-attention, allowing every current token to attend to every memory token through the standard attention mechanism (Dai et al., 2019; Xiao et al., 2025). Given input tokens $\mathbf{Z}_t \in \mathbb{R}^{N \times d}$ and memory tokens $\tilde{\mathbf{M}} \in \mathbb{R}^{M \times d}$, the effective context length is extended from $N$ to $N + M$:

$$\mathbf{Z}_t' = \text{Attn}\big([\tilde{\mathbf{M}}; \mathbf{Z}_t]\big)_{M:N+M} \tag{9}$$

where $[\cdot\,;\cdot]$ denotes concatenation along the sequence dimension and the subscript $M\colon N + M$ indicates that only the $N$ output tokens corresponding to the original input are retained after attention.

**Additive.** Memory representations are projected into the residual stream dimension and added directly to the input activations with a learnable vector $\boldsymbol{\alpha} \in \mathbb{R}^d$ scaling each dimension independently. Specifically, $\mathbf{W}_{\text{add}} \in \mathbb{R}^{d \times d_M}$ projects the memory from its encoding dimension $d_M$ into the residual stream dimension $d$, and $\alpha$ is initialized to a small value to ensure the memory representations do not overpower the residual stream early in training:

$$\mathbf{Z}_t' = \mathbf{Z}_t + \boldsymbol{\alpha} \odot \mathbf{W}_{\text{add}} \tilde{\mathbf{M}}, \tag{10}$$

where $\boldsymbol{\alpha} \in \mathbb{R}^d$ is a learnable vector and $\odot$ denotes elementwise multiplication.

**Cross Attention.** The current context queries the memory representations through a cross-attention layer applied after self-attention and before the feed-forward block. The input activations $\mathbf{Z}_t$ form the queries, while the memory $\tilde{\mathbf{M}}$ forms both the keys and values, allowing each current token to selectively retrieve information from the memory sequence:

$$\mathbf{Z}_t' = \mathbf{Z}_t + \text{Attn}\big(\mathbf{Z}_t \mathbf{W}_Q, \tilde{\mathbf{M}} \mathbf{W}_K, \tilde{\mathbf{M}} \mathbf{W}_V\big), \tag{11}$$

where $\mathbf{W}_Q, \mathbf{W}_K, \mathbf{W}_V \in \mathbb{R}^{d \times d_{\text{head}}}$ are learned projection matrices and the residual connection preserves the self-attention output.

**Adaptive Normalization (AdaNorm).** The memory representation conditions the layer normalization parameters via learned linear projections, scaling and shifting the normalized feature distribution to incorporate historical context (Dumoulin et al., 2017; Ho et al., 2022). Two projection matrices $\mathbf{W}_\gamma, \mathbf{W}_\beta \in \mathbb{R}^{d \times d_M}$ map the memory representation to per-channel scale and shift parameters:

$$\gamma = \mathbf{W}_\gamma \, \tilde{\mathbf{M}}, \quad \beta = \mathbf{W}_\beta \, \tilde{\mathbf{M}}, \tag{12}$$

$$\mathbf{Z}'_t = \gamma \odot \text{LayerNorm}(\mathbf{Z}_t) + \beta, \tag{13}$$

where $\odot$ denotes elementwise multiplication. This is analogous to conditional layer normalization used in diffusion models, with the memory representation playing the role of the conditioning signal.

**LoRA (QK Modulation).** Memory is injected into the query and key matrices through low-rank adaptation (Hu et al., 2022; Pouransari et al., 2025), altering the attention coefficients and, therefore, adjusting what the model attends to. For each of the query and key projections, a low-rank update is computed from the memory representation via two learned projection matrices $\mathbf{B} \in \mathbb{R}^{d \times r}$ and $\mathbf{A} \in \mathbb{R}^{r \times d_M}$, where $r \ll d$ is the rank:

$$\mathbf{Q}' = \mathbf{Q} + \mathbf{B}_Q \mathbf{A}_Q \tilde{\mathbf{M}}, \quad \mathbf{K}' = \mathbf{K} + \mathbf{B}_K \mathbf{A}_K \tilde{\mathbf{M}}. \tag{14}$$

## 5 Experimental Setup

### 5.1 World Model Architecture

Following (Zhou et al., 2025), our world model backbone is a standard Vision Transformer (ViT) (Dosovitskiy et al., 2021) that operates entirely in the latent space. We use a pretrained three-layer CNN encoder with stride of 2 (LeCun et al., 1989) and a VQ-VAE decoder with two layers of upsampling and a final layer for channel-wise projection (van den Oord et al., 2017). During training, the encoder is frozen and only the ViT predictor, memory mechanisms, and decoder are learned. The decoder serves visualization purposes only and its gradients do not flow through the predictor, following (Zhou et al., 2025).

We train the same ViT backbone for all combinations using four transformer layers, with memory injections integrated into layer at the designated injection points, shown in Figure 1. For SSM-based memory encoders, we use the Mamba state-space model (Gu and Dao, 2024) where the context is pre-encoded before passing to the transformer backbone.

All models are trained for 20 epochs with a window size of 10 and no overlap. We use the Adam optimizer with a learning rate of $3 \times 10^{-4}$, weight decay of $10^{-2}$, and cosine learning rate decay. To properly train the memory encoding methods that persist across windows, we truncate the gradients to be within the context window and carry over the detached hidden states to the next window, as described in (Williams and Peng, 1990; Sutskever, 2013)

### 5.2 Environment and Data

We evaluate each mechanism using the MemoryMaze dataset (Pasukonis et al., 2022), a memory-focused navigation benchmark that includes random wall colors and object placements throughout the environment, designed specifically to test memory recall. Following Deng et al. (2023), we generate scripted policies in a two-room environment where each episode varies in wall color and object placement, ensuring diverse perceptual conditions across trajectories. The dataset includes 29k episodes of 500 steps each.

Table 1: Comparison of encoder-injection pairings over a ten-step horizon. Best in **bold**, second in teal, third in orange. The baseline uses a context length of $C = 9$. We report each metric averaged over three seeds with standard errors.

| Encoding | Injection | SSIM ↑ | LPIPS ↓ | Image MSE ↓ | Lat. MSE ↓ | Cyc. MSE ↓ |
|----------|-----------|--------|---------|-------------|------------|------------|
| Baseline | None | $0.7026 \pm 0.023$ | $0.2852 \pm 0.0056$ | $0.06122 \pm 0.012$ | $1.089 \pm 0.15$ | $1.414 \pm 0.22$ |
| Cache | Prepend | $0.7355 \pm 0.043$ | $0.2664 \pm 0.039$ | $0.06282 \pm 0.027$ | $0.9091 \pm 0.057$ | $1.172 \pm 0.025$ |
| Cache | Additive | $0.698 \pm 0.023$ | $0.2909 \pm 0.0084$ | $0.06739 \pm 0.019$ | $1.096 \pm 0.077$ | $1.378 \pm 0.12$ |
| Cache | Cross Attn. | $0.6963 \pm 0.014$ | $0.2966 \pm 0.019$ | $0.06568 \pm 0.014$ | $1.202 \pm 0.12$ | $1.52 \pm 0.074$ |
| Cache | AdaNorm | $0.7037 \pm 0.027$ | $0.2843 \pm 0.007$ | $0.06486 \pm 0.013$ | $1.198 \pm 0.2$ | $1.533 \pm 0.21$ |
| Cache | LoRA | $0.6831 \pm 0.014$ | $0.2943 \pm 0.034$ | $0.05994 \pm 0.01$ | $1.167 \pm 0.23$ | $1.513 \pm 0.31$ |
| SSM | Prepend | $0.694 \pm 0.063$ | $0.3143 \pm 0.032$ | $0.06063 \pm 0.016$ | $1.085 \pm 0.17$ | $1.407 \pm 0.2$ |
| SSM | Additive | $0.6964 \pm 0.043$ | $0.3075 \pm 0.021$ | $0.07193 \pm 0.015$ | $1.144 \pm 0.18$ | $1.476 \pm 0.16$ |
| SSM | Cross Attn. | $0.6992 \pm 0.022$ | $0.279 \pm 0.027$ | $0.06434 \pm 0.009$ | $1.115 \pm 0.14$ | $1.393 \pm 0.12$ |
| SSM | AdaNorm | $0.6921 \pm 0.033$ | $0.2907 \pm 0.0082$ | $0.06203 \pm 0.0073$ | $1.025 \pm 0.17$ | $1.35 \pm 0.17$ |
| SSM | LoRA | $0.709 \pm 0.017$ | $0.274 \pm 0.011$ | $0.05278 \pm 0.0067$ | $1.12 \pm 0.21$ | $1.465 \pm 0.3$ |
| Titans | Prepend | $0.7098 \pm 0.027$ | $0.2739 \pm 0.018$ | $0.06956 \pm 0.0085$ | $1.109 \pm 0.18$ | $1.57 \pm 0.25$ |
| Titans | Additive | $0.6773 \pm 0.043$ | $0.3083 \pm 0.023$ | $0.0704 \pm 0.0096$ | $1.008 \pm 0.17$ | $1.271 \pm 0.2$ |
| Titans | Cross Attn. | $0.6703 \pm 0.023$ | $0.3467 \pm 0.031$ | $0.07248 \pm 0.025$ | $1.045 \pm 0.12$ | $1.44 \pm 0.18$ |
| Titans | AdaNorm | $0.6922 \pm 0.036$ | $0.3023 \pm 0.014$ | $0.06432 \pm 0.0094$ | $1.027 \pm 0.14$ | $1.324 \pm 0.17$ |
| Titans | LoRA | $0.7051 \pm 0.04$ | $0.2936 \pm 0.035$ | $0.05494 \pm 0.0085$ | $0.9555 \pm 0.12$ | $1.204 \pm 0.12$ |
| mLSTM | Prepend | $\mathbf{0.7834} \pm 0.018$ | $\mathbf{0.2085} \pm 0.018$ | $\mathbf{0.03174} \pm 0.013$ | $0.9042 \pm 0.094$ | $\mathbf{1.116} \pm 0.14$ |
| mLSTM | Additive | $0.7602 \pm 0.026$ | $0.2445 \pm 0.024$ | $0.04481 \pm 0.021$ | $\mathbf{0.8933} \pm 0.16$ | $1.173 \pm 0.19$ |
| mLSTM | Cross Attn. | $0.7103 \pm 0.025$ | $0.2699 \pm 0.022$ | $0.06713 \pm 0.016$ | $1.288 \pm 0.11$ | $1.651 \pm 0.13$ |
| mLSTM | AdaNorm | $0.7151 \pm 0.022$ | $0.2851 \pm 0.023$ | $0.05906 \pm 0.014$ | $1.154 \pm 0.23$ | $1.428 \pm 0.32$ |
| mLSTM | LoRA | $0.7338 \pm 0.023$ | $0.2712 \pm 0.033$ | $0.04537 \pm 0.014$ | $1.028 \pm 0.11$ | $1.287 \pm 0.18$ |

## 5.3 Evaluation Protocol

To assess memory capacity, we measure each memory-augmented world model's ability to recall past context over sequences of increasing horizon length $H$. Each episode consists of a *burn-in phase*, where the model processes a query sequence $\mathbf{X}_{t+1:t+H}$ given an initial context $\mathbf{X}_{t-C:t}$ ($C = 9$) to populate the memory mechanism and an *imagination phase*, where the model receives only the initial context and predicts the query sequence $\mathbf{X}_{t+1:t+H}$ fully in imagination, relying solely on its memory to recall the previously observed scenes. We evaluate horizon lengths $H \in \{10, 20, 50\}$ to characterize how memory recall degrades with increasing prediction distance.

We compare imagined rollouts with ground-truth sequences in both the image and latent spaces. For image reconstruction quality, we report SSIM (Wang et al., 2004), LPIPS (Zhang et al., 2018) with a pretrained VGG-16 backbone (Simonyan and Zisserman, 2014), and pixel-space MSE. In latent space, we measure the MSE between predicted and ground-truth representations. We additionally report "Cycle MSE", in which the predicted latent is decoded and re-encoded and the MSE is computed between this re-encoded latent and the ground truth. This metric serves as a diagnostic for latent-space collapse, where a low latent MSE may mask poor reconstructions by converging to a degenerate representation.

## 6 Results & Analysis

### 6.1 Recent Memory Recall

We first evaluate all encoder-injector combinations on a ten-step imagination horizon. Table 1 reports reconstruction quality and latent error metrics for all encoder-injection combinations.

The mLSTM-based memory encoder combined with context Prepend injection achieved the highest performance across all image reconstruction metrics and the lowest cycle-consistency error, while mLSTM with additive injection achieved the lowest latent mean squared error. Notably, the mLSTM encoder occupies the

top two positions across every metric, and the top three for MSE and latent MSE. This is unexpected given that the mLSTM compresses history into a fixed-size hidden state rather than retaining raw context frames as the cache does, which preserves full temporal detail. We expected the mLSTM to perform similarly to the SSM-based (Mamba) memory encoder due to their shared hidden state approach, however, mLSTM proves to outperform its SSM counterpart on all metrics. Two structural differences may explain this gap. First, the mLSTM's exponential forget gate $f_t = \exp(\tilde{f}_t)$ provides an explicit, input-dependent mechanism for discarding stale information. The retention weight assigned to a memory item written $k$ steps ago decays monotonically as $\prod_{j=1}^{k} f_{t-j+1}$, imposing a learnable recency bias. In contrast, the Mamba encoder achieves selectivity implicitly through input-dependent state transitions $\bar{\mathbf{A}}_t$, without an explicit gating mechanism that can sharply zero out outdated state components. Second, the mLSTM's matrix-valued cell state supports cross-channel correlations during retrieval, while Mamba's diagonal state does not (see Proposition 1 in Appendix C.1 for more details). We note that isolating the relative contribution of these two factors would require further ablation (e.g., equipping Mamba with exponential gating while retaining its diagonal structure), which we leave to future work.

The cache-based encoder with Prepend injection remains competitive, ranking third in SSIM, LPIPS, latent MSE, and second in cycle-consistency error, but is consistently surpassed by the mLSTM. We expected good performance from the cache-based encoder due to its direct access to uncompressed context frames.

The Titans memory encoders underperformed the baseline in most image reconstruction metrics but showed lower latent MSE, with the LoRA injection achieving the fourth lowest latent MSE among all configurations. The low latent error contrasting with poor image reconstruction quality exhibits signs of reconstruction collapse, as confirmed in the qualitative analysis discussed in Section 6.6. We identify two structural factors that may explain this underperformance. First, Titans' online gradient-based memory update introduces a second-order gradient dependency during outer-loop training, which is typically truncated in practice Williams and Peng (1990); Sutskever (2013), introducing bias into the predictor's gradient estimates (Appendix C.2). The other memory encoders avoid this issue through first-order recurrences or direct buffering. Second, Titans' surprise-based retention preserves memories according to their "surprise" or novelty rather than their temporal proximity, which is misaligned with the recency structure of next-frame prediction (see Appendix C.2 for further discussion). In contrast, mLSTM's exponential forget gate imposes a monotonically decaying retention bias that is structurally aligned with the predictive task.

Among the injection methods, the Prepend and additive approaches proved to be in the top-performing configurations across all metrics. The Prepend method pre-pends memory representations to the input context before self-attention, enabling joint processing of memory and input tokens across all attention heads. It is notable that the Cross-Attention injection method did not match the performance of Prepend, despite being a standard mechanism for integrating external context. We suspect this is due to a structural difference in how memory interacts with spatial features. In Prepend injection, memory tokens compete for attention with spatial features, providing a gating effect over spatial feature selection. For Cross-Attention injection, spatial self-attention is computed independently of the memory, and memory information is added as a post-hoc residual correction. This weakens memory tokens' influence over which spatial features the model attends to.

The Additive approach projects memory representations onto the residual stream with a learnable per-dimension scaling vector $\boldsymbol{\alpha}$ initialized to a small value (Section 4.2). This initialization creates a slowly growing memory contribution, allowing the predictor to first learn its dynamics function before integrating memory information. As the memory encoder matures, $\boldsymbol{\alpha}$ grows where the memory information is useful. This self-regulating mechanism prevents the memory from overpowering the self-attention outputs and explains the Additive method's robustness to injection depth (Section 6.2.2).

While the Prepend method performs as predicted, the strong performance of the additive approach, alongside the poor performance of cross-attention, challenges the assumption that all-token-channel methods outperform channel-wise alternatives. This suggests that the location of the memory injection in the transformer block may matter more than how many token channels it operates across.

## 6.2 Recall Across Multiple Horizons

To understand how memory mechanisms scale across long prediction horizons, we evaluate each combination at horizons $H \in \{10, 20, 50\}$. We isolate each design choice by fixing its counterpart to the best performing option based on Table 1. For instance, we compare the different memory encoding mechanisms by fixing the injection type to Prepend, which concatenates memory representations to the context without learned transformations before self-attention. Likewise, we compare injection mechanisms by using the mLSTM encoding method, which provides lossless access to past states and ensures that any differences are due to the injection mechanism alone.

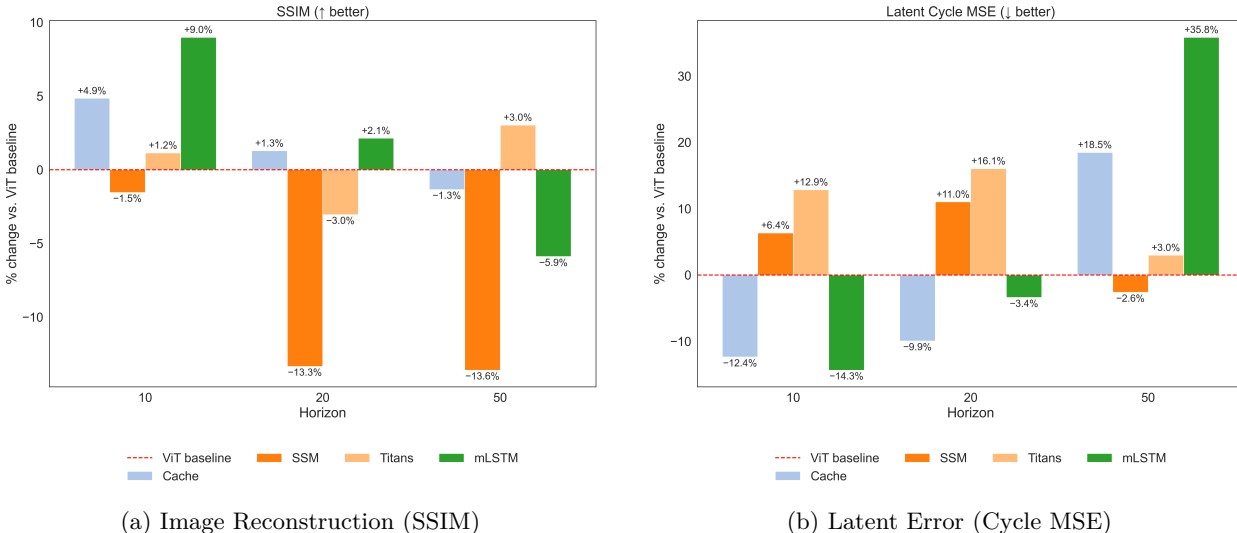

(a) Image Reconstruction (SSIM)  (b) Latent Error (Cycle MSE)

Figure 2: Encoder comparison across imagination horizons with a fixed Prepend injection. This isolates the effect of memory encoding by holding the injection mechanism constant. **(a)** SSIM and **(b)** Latent Cycle MSE for Cache, SSM, and Titans encoders reported as relative percentages with respect to the baseline ViT.

### 6.2.1 Encoder comparison (with Prepend injection).

Figure 2 compares the three memory encoders when paired with the Prepend injection across all horizon lengths. The mLSTM encoder leads in image reconstruction quality (SSIM) and latent error (Cycle MSE) at horizons 10 and 20, outperforming Cache, Titans, SSM and the baseline. However, this advantage does not hold at the longest 50 step horizon. mLSTM's latent error rises to the highest among all encoders ($\approx 2.29$), suggesting that its compressed hidden state becomes increasingly misaligned with the scene dynamics over extended rollouts, consistent with the qualitative hallucinations observed in Figure 7. The Cache encoder also beats the baseline for the 10 and 20 step horizons as expected, since the network has access to the raw states at inference time. It is interesting to note that mLSTM's compressed memory representations beat out the uncompressed Cache memory, suggesting that mLSTM's matrix memory learns to encode relevant details more effectively. In contrast, the other compressed memory encoder (SSM) failed to beat the baseline in any of the horizons. We hypothesize that this gap is due to the cross-channel representational capacity of mLSTM's matrix memory. As shown in Proposition 1 in Appendix C.1, the SSM encoder's diagonal state transition restricts its information retrieval to channel-specific operations. The mLSTM encoder, by contrast, produces a linear combination weighted by inner products across all stored channels, analogous to a linear attention operation over the memory history. This advantage is absent in the SSM's per-channel parameterization, regardless of how input-dependent gating is configured.

The Titans architecture stores memories in the weights of an MLP, giving it high theoretical capacity for cross-channel memory interactions. However, its surprise-based retention mechanism preserves memories according to their prediction error magnitude rather than temporal proximity. This can allow early but surprising memories to overpower the retention of recent information, which introduces a structural mismatch with the

recency prior of next-frame prediction. By contrast, mLSTM's exponential forget gate produces decaying retention weights, providing a learnable recency inductive bias that is aligned with the task (Equation 23).

### 6.2.2 Injection method comparison (with mLSTM encoding).

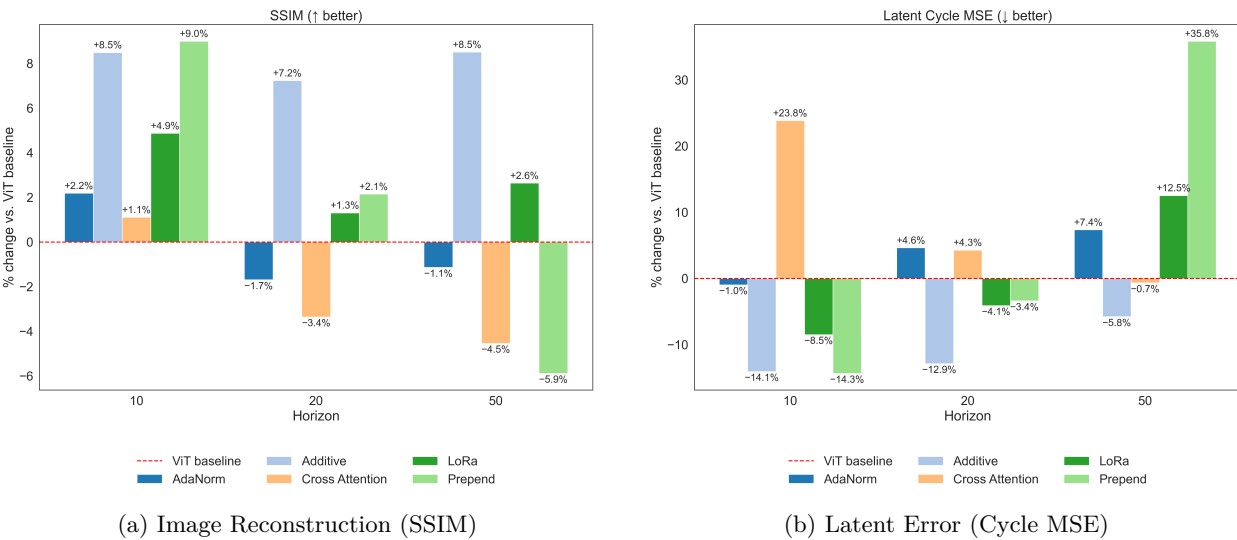

(a) Image Reconstruction (SSIM)    (b) Latent Error (Cycle MSE)

Figure 3: Injector comparison across imagination horizons with a fixed mLSTM encoder. This isolates the effect of the injection mechanism by holding the encoding constant. **(a)** SSIM and **(b)** Latent Cycle MSE for Prepend, Additive, Cross Attention, AdaNorm, and LoRA injectors reported as relative percentages with respect to the baseline ViT.

Figure 3 compares the five injection methods when paired with the mLSTM encoder across all horizon lengths. Prepend leads in image reconstruction at the 10 step horizon and also achieves the lowest latent error at that same horizon, making it the best short-horizon injection type. However, Prepend degrades as the horizons get longer, with latent error rising to the highest among the other injection methods while its image reconstruction quality falls near the baseline. This long-horizon fall in performance is consistent with the pattern observed in the encoder comparison (Section 6.2.1), and together suggest that the Prepend-mLSTM pairing is well-suited for short-to-medium horizon rollouts but accumulates latent drift under longer prediction horizons. Additive injection presents the most consistent performance profile across all horizons. At the shortest horizon it ranks second in both reconstruction quality and latent error. As the horizon gets longer it takes the lead in both reconstruction and latent error across 20 and 50 step horizons, beating the baseline and other injection methods. This consistent stability across horizons suggests that the learnable residual scaling in Additive injection effectively moderates the memory representation's influence on the residual stream, preventing the compounding latent drift observed with the Prepend method at longer horizons. Despite the mLSTM encoder providing strong memory representations, Cross-Attention yields the highest Latent Cycle MSE at the 10-step horizon, failing to beat the baseline. Similarly, the AdaNorm injection offers negligible improvement over the baseline in reconstruction quality and worse latent error on all horizons. LoRa occupies a middle position across short and medium horizons, with reconstruction and latent error just below that of Prepend and Additive. However, it degrades notably at the longest horizon, where its latent error rises well above the baseline, following Prepend. Overall, the Additive injection emerges as the most robust choice across all horizons when the memory encoder quality is high, while Prepend offers a strong short-horizon alternative at the cost of long-horizon latent stability.

By default, memory is injected at every transformer layer. To assess whether this is necessary, we test the application of the injection method on all layers, on the first layer, and on the middle layer separately within the four-layer ViT predictor. We select the two top-performing injection methods from Section 6.1, (mLSTM + Additive and mLSTM + Prepend) and evaluate each at all three depths across rollout horizons of 10, 20, and 50 steps, shown in Figure 4.

### 6.3 Ablating Injection Depth

For Additive injection, all three depth positions remain competitive across all horizons in both reconstruction quality (SSIM) and latent error (Cycle MSE), with the ranking shifting across horizons. At the shortest 10 step horizon, Additive-middle achieves the highest SSIM and the lowest latent error among all six combinations, while Additive-all follows closely. By the 20 step horizon, Additive-all leads in both metrics, with Additive-first and Additive-middle trailing only slightly. At the longest horizon, Additive-middle exhibits the lowest latent error of all combinations. We note here that no Additive depth position fails significantly at any horizon, suggesting that Additive injection is robust to depth placement and that its residual scaling mechanism functions effectively regardless of where in the network the memory is introduced.

The Prepend injection results tells a markedly different story. Prepend-all remains competitive in SSIM at the 10 step horizon, tracking closely with the Additive combinations, and its latent error is the lowest among the three Prepend variants. However, Prepend-first and Prepend-middle both exhibit high latent error for all three horizons, with values far above any Additive combination. Notably, Prepend-first and Prepend-middle's reconstruction values appear moderate at shorter horizons relative to their latent errors, indicating that these combinations produce structurally plausible reconstructions but are inconsistent with the input scene in the latent space.

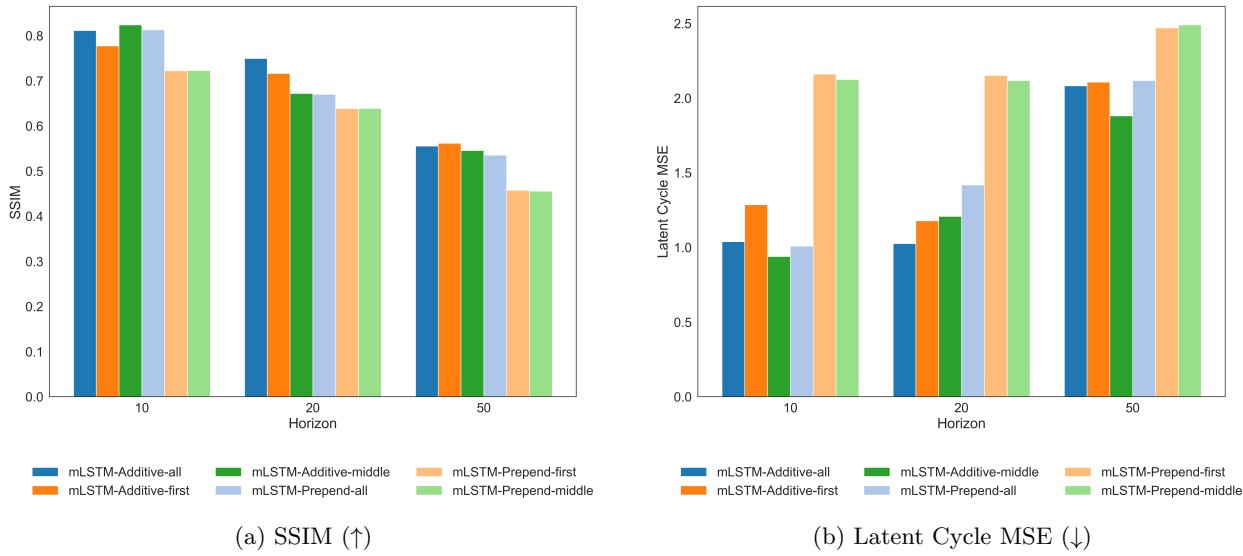

(a) SSIM (↑)                                         (b) Latent Cycle MSE (↓)

Figure 4: Ablating injection depth for mLSTM Additive and Prepend injections across rollout horizons. (a) Image reconstruction quality (SSIM). (b) Latent error (Cycle MSE).

The contrast between Prepend-all and the restricted Prepend variants reveals that the full-depth application of Prepend injection is a necessary condition for latent stability. When Prepend is applied at only a subset of layers (Prepend-first), memory information must be propagated through subsequent layers without reinforcement. Subsequent layers' attention patterns are learned without memory context and may overwrite the memory signal in the residual stream. Prepend-all avoids this by providing memory keys at every self-attention computation, giving the network several opportunities to route memory information into the residual stream rather than relying on a single injection point. In contrast, Additive injection writes memory directly to the residual stream via a learned projection, making the memory signal less dependent on attention patterns.

Taken together, these results indicate that injection depth interacts more critically with Prepend than with Additive injection. Additive injection is robust across all depth configurations, making it a practical choice when full-layer injection is computationally limiting. Prepend injection, while capable of strong performance when applied at all layers, degrades severely under depth restriction.

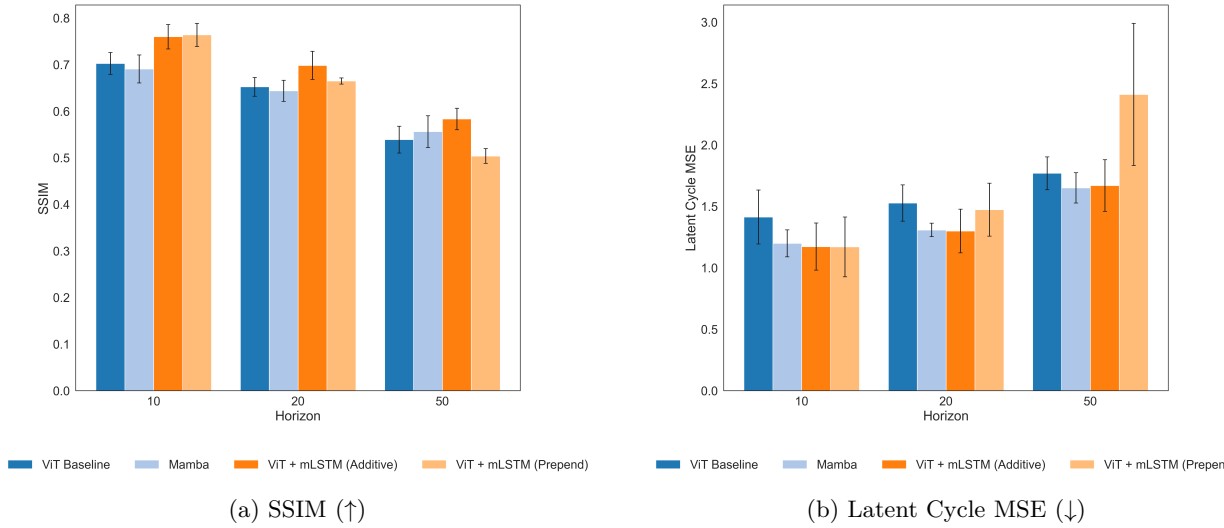

(a) SSIM (↑)    (b) Latent Cycle MSE (↓)

Figure 5: Backbone comparison across imagination horizons. (a) Image reconstruction quality (SSIM). (b) Latent error (Cycle MSE).

## 6.4 Comparing ViT+Memory vs. Pure Mamba Backbones

Prior work has shown that pure SSM backbones achieve strong long-horizon recall (Deng et al., 2023; Samsami et al., 2024; Po et al., 2025; Savov et al., 2025; Lee et al., 2025). We explore whether a hybrid, short-context ViT augmented with memory mechanisms can match a pure Mamba backbone. We train a Mamba backbone of comparable parameter count on the same task and compare against both the baseline ViT and the top two memory-augmented ViT combination (mLSTM + Additive and mLSTM + Prepend) across rollout horizons of 10, 20, and 50 steps. Results are shown in Figure 5.

The pure Mamba backbone fails to improve over the ViT Baseline in reconstruction quality at any horizon, tracking at or slightly below it in SSIM across all three evaluation points. This is a notable negative result given the strong long-horizon recall attributed to SSM-based architectures in prior work. However, Mamba does achieve consistently lower latent error than the baseline across all horizons, indicating that its hidden state representations maintain consistency even when its reconstruction quality doesn't improve. This separation in reconstruction quality and latent consistency in the pure Mamba backbone mirrors the pattern observed for certain injection methods in Section 6.2.2, and suggests that the Mamba backbone learns more latently coherent representations while sacrificing perceptual sharpness.

Both memory-augmented combinations outperform the ViT Baseline and pure Mamba in reconstruction quality at the 10 and 20 step horizons. mLSTM Additive injection is the most consistent of the four configurations, leading in reconstruction quality at horizons 10 and 20 while matching Mamba for the lowest latent error. In the 50 step horizon, mLSTM Additive exhibits the highest reconstruction quality and achieves latent error comparable to Mamba, demonstrating that the Additive injection can match the latent stability of the Mamba backbone while maintaining perceptual quality.

mLSTM with Prepend injection matches Additive in reconstruction quality at the shortest horizon but diverges sharply in latent error at the longest horizon, where it exceeds all other combinations including the ViT baseline. This shows that the longer-horizon instability of Prepend injection method identified in Section 6.2.2 persists and further emphasizes that reconstruction quality alone is insufficient to compare memory mechanisms.

These results indicate that a carefully chosen memory mechanism used to augment a ViT can match or exceed a pure Mamba backbone of comparable size across prediction horizons, without requiring a full architectural replacement. The Additive injection combination in particular achieves the best overall tradeoff between reconstruction quality and latent consistency, suggesting that lightweight memory augmentation of

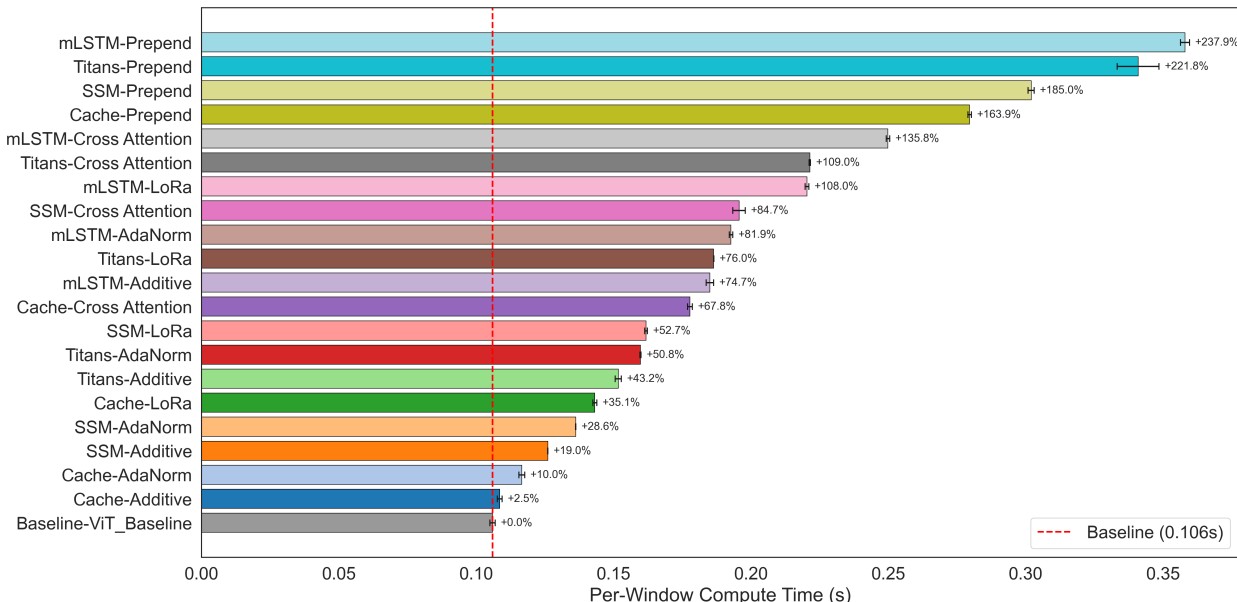

Figure 6: Per-window training compute time for all encoder-injection combinations. The dashed red line indicates the ViT baseline (0.106s). Percentages denote overhead relative to the baseline.

an existing ViT predictor is a viable and effective alternative to adopting a recurrent backbone for long-horizon scene prediction.

## 6.5  Computational Cost Analysis

We compare the per-window training compute time across all encoder-injection combinations relative to the ViT baseline. For each combination, we report the mean compute time averaged over three seeds along with the relative overhead. Results are visualized in Figure 6.

We observe that compute overhead is driven primarily by the injection method rather than the encoding method. Prepend injection is the most expensive across all encoders, ranging from +164% (Cache) to +238% (mLSTM), since it increases the sequence length of every self-attention operation. As expected, Cross attention is the second most costly (+68% to +136%), as it introduces an additional attention operation per layer. By contrast, the Additive injection adds minimal overhead (+2.5% for Cache and +19% for SSM), since it requires only an elementwise addition to the residual stream. AdaNorm and LoRA fall in between, with LoRA generally more expensive than AdaNorm due to the additional low-rank projections. Within each injection method, memory encoding cost increases from Cache (cheapest, as it stores raw frames with no encoding forward pass) through SSM and Titans to the most expensive mLSTM.

mLSTM with Additive injection is the best-performing configuration from Section 6.2.2 which incurs +74.7% overhead (0.185s per window), placing it in the middle of the cost distribution. This represents a favorable quality-cost tradeoff given that it achieves the strongest reconstruction and latent error values while costing less than half the compute of any Prepend configuration. The mLSTM with Prepend injection, which also performed well, is the single most expensive configuration at +238%. These results suggest that the Additive injection should be preferred in practice, as it captures most of the benefit of the mLSTM encoder at substantially lower cost than attention-based injection alternatives.

### 6.6 Qualitative Analysis

In Figure 7, we show a 20 step imagination horizon for the ViT Baseline, pure Mamba baseline, and select encoder-injection combinations for each memory encoding type. The ground truth rollout is shown in the top row with a green border.

Most combinations maintain plausible scene structure at short horizons ($\leq 10$ frames), correctly rendering the geometry and object positions. The differences become most apparent in the mid-to-late frames, where wall color and surface texture begin to diverge from the ground truth. This qualitative degradation is consistent with the rise in latent cycle error observed at longer horizons across all configurations in the quantitative results.

The weaker configurations, particularly Titans with Prepend and SSM with LoRA, exhibit early color hallucination, with wall hues shifting to incorrect colors and background textures losing coherence well before the final frames. mLSTM with the Additive injection maintains more consistent scene colors and structural boundaries through the mid-horizon frames, consistent with its lower latent cycle error at the 20 step horizon.

The pure Mamba backbone produces coherent rollouts in early frames but exhibits significant color drift in later frames, eventually converging toward a uniform appearance. We note that Mamba processes a flattened sequence of spatial patches without explicit 2D structure, whereas the ViT's self-attention jointly processes all spatial patches with position-aware interactions. This architectural difference means Mamba lacks the spatial inductive bias that allows the ViT to maintain fine-grained spatial coherence across patches during decoding, potentially explaining the gap between Mamba's coherent latent dynamics and its weaker reconstruction quality observed in Section 6.4. Investigating whether 2D-aware scanning strategies (e.g., bidirectional or zigzag scans) can improve Mamba's reconstruction fidelity is left to future work.

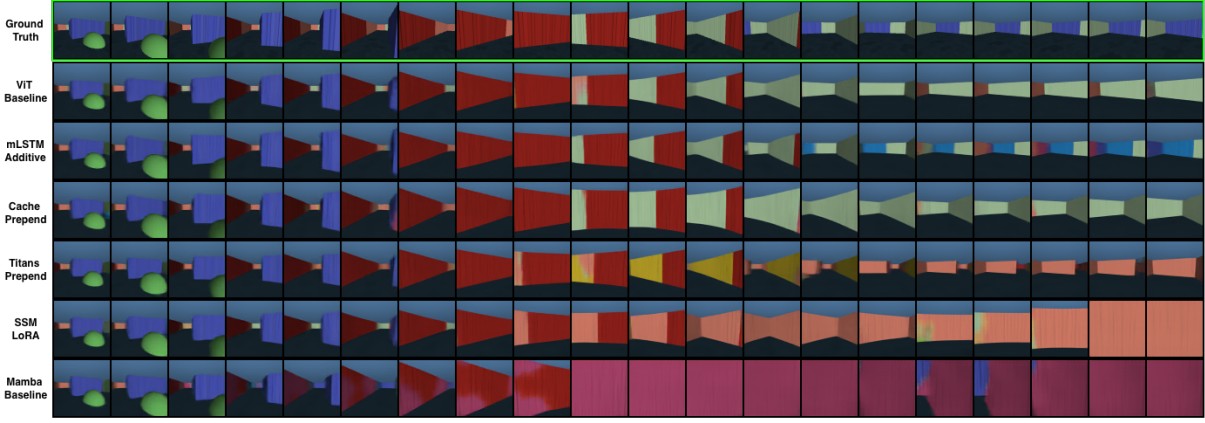

Figure 7: Visual comparison of 20 imagined steps in the MemoryMaze environment for the ViT baseline, Mamba baseline, and selected encoder-injection pairs for each encoder type. Top row (green): ground truth.

## 7 Discussion

**Practical guidance.** Our results suggest that a mLSTM encoding with the Additive injection at all transformer layers offers the best balance of recall quality and computational cost. It achieves the lowest latent error values and near-best reconstruction quality at +75% compute overhead with less than half the cost of any Prepend configuration. When maximum reconstruction fidelity is required regardless of cost, mLSTM with Prepend injection provides the highest SSIM and lowest MSE, but at +238% overhead. Cache-based encoding remains a reasonable baseline when implementation simplicity is prioritized, but does not justify its conceptual appeal as an "uncompressed" memory.

**Memory span vs. compression vs. compute.** The encoder-injector taxonomy reveals a fundamental tension in memory design. Cache encoders preserve full fidelity but grow linearly, creating a hard upper bound on temporal span. SSM and mLSTM encoders compress the entire history into a fixed representation, enabling unbounded span at the cost of lossy compression, though the mLSTM's matrix state substantially mitigates this loss. Titans encoders attempt adaptive compression through online learning, but the resulting training instability limits their practical utility in the world modeling setting. On the injection side, methods that expand the sequence length, such as Prepend, offer expressive memory processing but incur quadratic cost in the attention computation and are sensitive to injection depth, while additive injection is cheap, robust to depth restriction, and effective when paired with a strong encoder.

**Limitations and Future Directions** A limitation of our study is that we evaluate a single environment(MemoryMaze two-room) and a single task (recent-state recall). We use a single controlled environment deliberately: MemoryMaze is a standard memory-recall benchmark whose randomized wall colors and object placements are designed to stress recall, and holding it fixed lets us attribute differences to the memory mechanism rather than to environment-specific factors. It nonetheless leaves open how the rankings transfer to richer dynamics, larger scenes, and other benchmarks such as BlockWorld Lillemark et al. (2026), and to downstream tasks such as reinforcement learning or model-predictive control-based planning. Second, our world model backbone is a relatively small four-layer ViT; scaling behavior with larger models and longer sequences remains unexplored. Finally, the memory-selection axis discussed briefly in Section 3.3 is a natural next direction. We expect learned selection to benefit cache- and retrieval-style mechanisms most, where the buffer is finite, and leave a controlled study of selection policies to future work.

## 8 Conclusion

We investigated memory mechanisms for extending the effective memory span of transformer-based world models. By decomposing memory into encoding and injection components, we evaluated twenty encoder-injector combinations on the MemoryMaze benchmark across multiple imagination horizons.

Our central finding is that the mLSTM memory encoder, which compresses history into a matrix-valued hidden state, consistently outperforms all alternatives across both image reconstruction quality and latent consistency metrics. When paired with the Additive injection method at all transformer layers, it achieves the strongest overall performance at moderate computational cost. This result challenges the intuition that lossless memory access should dominate, and suggests that learned, selective compression can be more effective than raw storage when the state representation is sufficiently expressive. A comparison against a pure Mamba backbone demonstrates that the memory-augmented ViT matches or slightly exceeds the Mamba baseline while retaining architectural modularity.

These results establish a practical framework for augmenting transformer-based world models with memory. Future work should evaluate these mechanisms on downstream reinforcement learning tasks, larger model scales, and environments requiring explicit loop closure, where the mLSTM's selective memory gating may prove particularly advantageous.

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

# A  Additional Experiment Details

This section provides additional experimental details beyond what is presented in Section 5, organized into architecture, training, evaluation, and hardware/compute.

## A.1  Architecture Details

**Visual encoder.**  We use a three-layer ResNet-style CNN encoder operating at $64 \times 64$ input resolution. The encoder has stem width 32, two residual stages of two blocks each with channel widths $[32, 64, 128]$, and strides $[2, 2, 2]$, producing tokens of embedding dimension $d = 256$. We use GroupNorm with 32 groups throughout. The encoder is pretrained and frozen during predictor training, and its weights are shared across all encoder-injection combinations.

**Decoder.**  The decoder follows the small VQ-VAE architecture of van den Oord et al. (2017), with quantization disabled so that the decoder operates on continuous latents. The decoder uses 128 channels, four residual blocks (128 channels each), stride 2 upsampling, and an embedding dimension of 256 matching the encoder. Decoder gradients do not flow through the predictor, following Zhou et al. (2025). We use smooth $L_1$ reconstruction loss.

**ViT predictor.**  The predictor is a four-layer Vision Transformer with hidden dimension $d = 256$, 16 attention heads, MLP expansion dimension 2048, and dropout 0.1 applied after attention and MLP sub-blocks. The context length is $C = 9$ history frames with 1 prediction frame per window. All memory injections are applied at every transformer layer by default (see Section 6.2.2 for the depth ablation), with two exceptions: Prepend operates on the input sequence before the first self-attention, and AdaNorm modulates the layer normalization between the self-attention and feed-forward sub-blocks.

**Memory encoder dimensions.**  All memory encoders operate at the predictor's residual stream dimension $d = 256$, with the following configuration choices:

- **Cache :** buffer size $K = 10$, matching the training window size.

- **Titans  (Behrouz et al., 2024):** inner-loop MLP over a 512-dimensional memory state with a single hidden layer; the online update uses a per-parameter learned forget gate.

- **SSM  (Gu and Dao, 2024):** a single Mamba layer with state dimension 512 and selective-scan parameterization (rank 128 for the $\Delta$ projection). The context is pre-encoded by the Mamba layer before being passed to the transformer backbone.

- **mLSTM  (Beck et al., 2024):** a matrix memory cell held *per patch* and split across $H = 8$ heads of head-dimension $d_h = 32$ (i.e., eight $32 \times 32$ blocks per patch rather than a single $256 \times 256$ matrix), with exponential input and forget gates as defined in Section 4.1..

**LoRA injection parameters.**  When used as an injection method, LoRA QK modulation (Hu et al., 2022) (Section 4.2) uses rank $r = 16$ and scaling factor $\alpha = 0.5$, applied independently to the query and key projections at every transformer layer.

**Action and proprioceptive conditioning.** Discrete actions are embedded via a learned 16-dimensional embedding and projected to a 12-dimensional conditioning vector. Proprioceptive observations (agent pose, 4 channels) are embedded through a small MLP to the same 12-dimensional conditioning space. Action and proprioceptive embeddings are concatenated along the token dimension before being passed to the predictor.

**Mamba backbone baseline.** The pure Mamba backbone (Gu and Dao, 2024) used in Section 6.4 consists of stacked Mamba blocks with parameter count comparable to the ViT predictor. The backbone operates on flattened spatial patches without explicit 2D position encoding, which we discuss as a possible source of its weaker reconstruction fidelity in Section 6.6.

### A.2 Training Details

**Dataset.** We train on the MemoryMaze 3x7 two-room dataset generated following Deng et al. (2023), split 90/10/10 between training, validation, and test set. Episodes are 500 frames long and are processed in chunks of 10 frames with no overlap between training windows.

**Optimizer and schedule.** We use Adam with $\beta_1 = 0.9$, $\beta_2 = 0.999$. Learning rates are set per-module: predictor $3 \times 10^{-4}$, encoder and decoder $1 \times 10^{-3}$, action encoder $3 \times 10^{-4}$. Weight decay is 0.1 for the predictor and 0.01 for all other modules. Gradient norm is clipped at 1.0. We use a cosine annealing schedule with warm restarts ($T_0 = 40$, $T_{\text{mult}} = 2$, $\eta_{\min}$ ratio $= 0.1$) and a 1% linear warm-up. All models are trained for 20 epochs with effective batch size 128.

**Gradient truncation across windows.** Gradients are truncated to within each training window, following the truncated backpropagation through time protocol of Williams and Peng (1990); Sutskever (2013). For encoders with persistent state across windows (Titans, SSM, mLSTM), the hidden state is detached at the end of each window and carried forward to the next. The Cache encoder buffer is similarly carried across windows without gradient flow.

### A.3 Evaluation Details

At evaluation time, we sample trajectories of 250 frames from held-out episodes. Each trajectory is divided into a burn-in phase of length $C = 9$ and an imagination phase of length $H \in \{10, 20, 50\}$. During burn-in, the model receives ground-truth observations and updates its memory state. During imagination, the model receives only its own predicted latents as input and rolls out autoregressively for $H$ steps.

All reported metrics (SSIM, LPIPS, image MSE, latent MSE, cycle MSE) are computed per-frame against the corresponding ground-truth frame and averaged over the imagination horizon. For cycle MSE, the predicted latent is passed through the frozen decoder and then re-encoded through the frozen encoder before comparison with the ground-truth latent. Each configuration is trained with three random seeds $\{0, 1, 2\}$; we report the mean across seeds with the standard error of the mean ($\sigma/\sqrt{3}$).

### A.4 Hardware and Compute

All experiments were conducted on NVIDIA A100 80GB GPUs. The per-window compute measurements reported in Figure 6 were collected on a single A100 to isolate the relative cost of each combination from multi-GPU communication overhead.

## B Extended Imagination Horizons

We report the same encoder-injection comparison over longer imagination horizons of twenty and fifty steps in Tables 2 and 3. These results characterize how each combination degrades as the predictor is required to roll out further beyond its context window.

**Twenty-step horizon.** The results from Section 6.1 carries over: mLSTM with additive injection takes the top SSIM, LPIPS, and cycle-consistency error, and mLSTM with Prepend injection retains the lowest

Table 2: Comparison of encoder-injection pairings over a twenty-step horizon. Best in **bold**, second in teal, third in orange. The baseline uses a context length of $C = 9$. We report each metric averaged over three seeds with standard errors.

| Encoding | Injection | SSIM ↑ | LPIPS ↓ | Image MSE ↓ | Lat. MSE ↓ | Cyc. MSE ↓ |
|---|---|---|---|---|---|---|
| Baseline | None | $0.6525 \pm 0.02$ | $0.309 \pm 0.025$ | $0.06666 \pm 0.011$ | $1.178 \pm 0.08$ | $1.528 \pm 0.15$ |
| Cache | Prepend | $0.6618 \pm 0.043$ | $0.3143 \pm 0.055$ | $0.07451 \pm 0.031$ | $\mathbf{1.007} \pm 0.079$ | $1.345 \pm 0.1$ |
| Cache | Additive | $0.6023 \pm 0.07$ | $0.3517 \pm 0.054$ | $0.0965 \pm 0.039$ | $1.172 \pm 0.16$ | $1.451 \pm 0.15$ |
| Cache | Cross Attn. | $0.65 \pm 0.05$ | $0.3349 \pm 0.05$ | $0.07771 \pm 0.017$ | $1.336 \pm 0.26$ | $1.724 \pm 0.26$ |
| Cache | AdaNorm | $0.578 \pm 0.022$ | $0.4047 \pm 0.024$ | $0.08727 \pm 0.017$ | $1.348 \pm 0.11$ | $1.674 \pm 0.12$ |
| Cache | LoRA | $0.6127 \pm 0.017$ | $0.3459 \pm 0.026$ | $0.07135 \pm 0.0049$ | $1.261 \pm 0.16$ | $1.604 \pm 0.23$ |
| SSM | Prepend | $0.5662 \pm 0.04$ | $0.4251 \pm 0.033$ | $0.09094 \pm 0.014$ | $1.348 \pm 0.24$ | $1.665 \pm 0.21$ |
| SSM | Additive | $0.5443 \pm 0.0043$ | $0.4467 \pm 0.013$ | $0.1196 \pm 0.019$ | $1.352 \pm 0.14$ | $2.011 \pm 0.19$ |
| SSM | Cross Attn. | $0.6344 \pm 0.04$ | $0.3697 \pm 0.073$ | $0.08349 \pm 0.0085$ | $1.319 \pm 0.019$ | $1.582 \pm 0.038$ |
| SSM | AdaNorm | $0.609 \pm 0.028$ | $0.3572 \pm 0.026$ | $0.07949 \pm 0.012$ | $1.193 \pm 0.11$ | $1.573 \pm 0.13$ |
| SSM | LoRA | $0.6435 \pm 0.024$ | $0.317 \pm 0.026$ | $0.06311 \pm 0.0045$ | $1.228 \pm 0.13$ | $1.584 \pm 0.21$ |
| Titans | Prepend | $0.6333 \pm 0.035$ | $0.332 \pm 0.039$ | $0.08201 \pm 0.015$ | $1.216 \pm 0.13$ | $1.748 \pm 0.22$ |
| Titans | Additive | $0.6059 \pm 0.059$ | $0.3833 \pm 0.038$ | $0.08921 \pm 0.02$ | $1.098 \pm 0.079$ | $1.408 \pm 0.12$ |
| Titans | Cross Attn. | $0.5854 \pm 0.035$ | $0.4379 \pm 0.047$ | $0.1046 \pm 0.027$ | $1.271 \pm 0.12$ | $1.701 \pm 0.11$ |
| Titans | AdaNorm | $0.6032 \pm 0.089$ | $0.3832 \pm 0.076$ | $0.09021 \pm 0.03$ | $1.128 \pm 0.13$ | $1.455 \pm 0.15$ |
| Titans | LoRA | $0.6176 \pm 0.065$ | $0.3622 \pm 0.07$ | $0.09257 \pm 0.034$ | $1.074 \pm 0.12$ | $1.531 \pm 0.22$ |
| mLSTM | Prepend | $0.6649 \pm 0.0065$ | $0.2779 \pm 0.017$ | $\mathbf{0.05283} \pm 0.014$ | $1.176 \pm 0.2$ | $1.474 \pm 0.22$ |
| mLSTM | Additive | $\mathbf{0.6985} \pm 0.03$ | $\mathbf{0.2619} \pm 0.021$ | $0.06132 \pm 0.023$ | $1.009 \pm 0.16$ | $\mathbf{1.3} \pm 0.18$ |
| mLSTM | Cross Attn. | $0.6322 \pm 0.046$ | $0.3318 \pm 0.055$ | $0.0876 \pm 0.029$ | $1.199 \pm 0.15$ | $1.562 \pm 0.14$ |
| mLSTM | AdaNorm | $0.6385 \pm 0.035$ | $0.3152 \pm 0.017$ | $0.08354 \pm 0.024$ | $1.336 \pm 0.28$ | $1.633 \pm 0.32$ |
| mLSTM | LoRA | $0.6597 \pm 0.0064$ | $0.3294 \pm 0.0039$ | $0.05928 \pm 0.0093$ | $1.166 \pm 0.035$ | $1.458 \pm 0.1$ |

image MSE. Latent MSE is the one metric where mLSTM is displaced. Cache+Prepend achieves the lowest value ($1.007 \pm 0.079$), narrowly edging mLSTM+Additive ($1.009 \pm 0.16$). This is consistent with the h=10 behavior, where Cache+Prepend also ranked among the top three on latent error: direct buffering of uncompressed context frames remains a strong inductive bias for latent prediction when the horizon is moderate. The ordering among injection methods within the mLSTM family also begins to separate. Additive pulls ahead of Prepend on four of five metrics, compared to the near-tie at h=10. Finally, the fraction of combinations beating the baseline drops sharply at h=20 (e.g., only 2/20 on LPIPS versus 10/20 at h=10), indicating that weaker memory mechanisms quickly lose their advantage over the context-9 ViT once rollout length exceeds the training horizon.

**Fifty-step horizon.** At h=50, mLSTM+Additive continues to win the three reconstruction metrics (SSIM, LPIPS, and image MSE), now by a larger margin than at shorter horizons. However, the latent-space metrics are now led by the Titans combinations: Titans+LoRA achieves the lowest latent MSE ($1.205 \pm 0.12$) and Titans+Additive achieves the lowest cycle-consistency error ($1.531 \pm 0.19$), with three of the top four positions on both metrics occupied by Titans configurations. Titans' persistent, online-updated memory appears to provide a benefit at long horizons that is not visible at h=10, where its second-order gradient bias and surprise-based retention (Section 6.1) dominate the picture. The gap is limited to latent space, though, Titans continues to underperform mLSTM on pixel-level reconstruction, consistent with the reconstruction collapse we observe qualitatively (Section 6.6). SSM+LoRA emerges as a strong generalist at h=50, ranking second on SSIM and image MSE and third on LPIPS, matching the pattern observed at h=10 where LoRA was the SSM family's best injection. In contrast, SSM+Additive collapses across all metrics as the horizon grows (SSIM drops from 0.696 at h=10 to 0.320 at h=50; cycle MSE grows from 1.476 to 3.175), making it the worst combination at h=50 by a wide margin.

Overall, the long-horizon results support the two main conclusions of Section 6.1: the mLSTM encoder dominates pixel-space reconstruction across all tested horizons, and the Additive and Prepend injection methods remain the strongest choices. The one new observation is that Titans becomes competitive on

Table 3: Comparison of encoder-injection pairings over a fifty-step horizon. Best in **bold**, second in teal, third in orange. The baseline uses a context length of $C = 9$. We report each metric averaged over three seeds with standard errors.

| Encoding | Injection | SSIM ↑ | LPIPS ↓ | Image MSE ↓ | Lat. MSE ↓ | Cyc. MSE ↓ |
|---|---|---|---|---|---|---|
| Baseline | None | $0.5391 \pm 0.029$ | $0.4226 \pm 0.027$ | $0.1022 \pm 0.013$ | $1.355 \pm 0.11$ | $1.77 \pm 0.13$ |
| Cache | Prepend | $0.5227 \pm 0.058$ | $0.4795 \pm 0.087$ | $0.1187 \pm 0.036$ | $1.555 \pm 0.26$ | $2.059 \pm 0.34$ |
| Cache | Additive | $0.4703 \pm 0.01$ | $0.4933 \pm 0.023$ | $0.149 \pm 0.013$ | $1.486 \pm 0.24$ | $1.803 \pm 0.34$ |
| Cache | Cross Attn. | $0.5254 \pm 0.061$ | $0.4577 \pm 0.078$ | $0.1035 \pm 0.028$ | $1.351 \pm 0.25$ | $1.611 \pm 0.28$ |
| Cache | AdaNorm | $0.4182 \pm 0.1$ | $0.5268 \pm 0.1$ | $0.1815 \pm 0.06$ | $1.938 \pm 0.31$ | $2.658 \pm 0.63$ |
| Cache | LoRA | $0.5292 \pm 0.036$ | $0.4275 \pm 0.028$ | $0.1082 \pm 0.013$ | $1.49 \pm 0.2$ | $1.895 \pm 0.27$ |
| SSM | Prepend | $0.4635 \pm 0.021$ | $0.5073 \pm 0.019$ | $0.1256 \pm 0.013$ | $1.368 \pm 0.29$ | $1.729 \pm 0.24$ |
| SSM | Additive | $0.3196 \pm 0.032$ | $0.6618 \pm 0.031$ | $0.2249 \pm 0.039$ | $1.55 \pm 0.26$ | $3.175 \pm 0.36$ |
| SSM | Cross Attn. | $0.5435 \pm 0.097$ | $0.4427 \pm 0.13$ | $0.1159 \pm 0.047$ | $1.354 \pm 0.16$ | $1.739 \pm 0.23$ |
| SSM | AdaNorm | $0.5093 \pm 0.048$ | $0.4689 \pm 0.057$ | $0.1227 \pm 0.014$ | $1.667 \pm 0.26$ | $2.147 \pm 0.27$ |
| SSM | LoRA | $0.5833 \pm 0.017$ | $0.4168 \pm 0.032$ | $0.08751 \pm 0.0099$ | $1.364 \pm 0.14$ | $1.758 \pm 0.23$ |
| Titans | Prepend | $0.5499 \pm 0.046$ | $0.427 \pm 0.059$ | $0.09906 \pm 0.011$ | $1.396 \pm 0.27$ | $1.831 \pm 0.37$ |
| Titans | Additive | $0.5677 \pm 0.031$ | $0.4357 \pm 0.026$ | $0.09925 \pm 0.013$ | $1.217 \pm 0.15$ | $\mathbf{1.531} \pm 0.19$ |
| Titans | Cross Attn. | $0.5036 \pm 0.051$ | $0.5239 \pm 0.068$ | $0.1481 \pm 0.046$ | $1.409 \pm 0.17$ | $1.946 \pm 0.24$ |
| Titans | AdaNorm | $0.4998 \pm 0.086$ | $0.4596 \pm 0.081$ | $0.134 \pm 0.034$ | $1.221 \pm 0.22$ | $1.612 \pm 0.31$ |
| Titans | LoRA | $0.5054 \pm 0.067$ | $0.4882 \pm 0.075$ | $0.1291 \pm 0.034$ | $\mathbf{1.205} \pm 0.12$ | $1.695 \pm 0.12$ |
| mLSTM | Prepend | $0.504 \pm 0.016$ | $0.4724 \pm 0.025$ | $0.1346 \pm 0.022$ | $1.785 \pm 0.3$ | $2.412 \pm 0.58$ |
| mLSTM | Additive | $\mathbf{0.5835} \pm 0.023$ | $\mathbf{0.3675} \pm 0.02$ | $\mathbf{0.08144} \pm 0.002$ | $1.295 \pm 0.18$ | $1.67 \pm 0.21$ |
| mLSTM | Cross Attn. | $0.5057 \pm 0.062$ | $0.4271 \pm 0.072$ | $0.12 \pm 0.019$ | $1.329 \pm 0.25$ | $1.767 \pm 0.32$ |
| mLSTM | AdaNorm | $0.5334 \pm 0.033$ | $0.4282 \pm 0.021$ | $0.1185 \pm 0.019$ | $1.599 \pm 0.24$ | $1.91 \pm 0.3$ |
| mLSTM | LoRA | $0.5515 \pm 0.052$ | $0.4012 \pm 0.046$ | $0.107 \pm 0.016$ | $1.61 \pm 0.26$ | $2.014 \pm 0.38$ |

Table 4: Effect of enlarging the SSM recurrent state ($S{=}512{\to}2048$, $\Delta$-rank $\to 512$) at fixed Prepend and Additive injectors.

| Encoding / Injection | $S$ | SSIM ↑ | | | Cycle MSE ↓ | | |
|---|---|---|---|---|---|---|---|
| | | h=10 | h=20 | h=50 | h=10 | h=20 | h=50 |
| SSM + Prepend | 512 | 0.748 | 0.584 | 0.505 | 1.515 | 1.955 | 2.190 |
| SSM + Prepend | 2048 | 0.713 | 0.601 | 0.412 | 1.721 | 1.726 | 2.429 |
| SSM + Additive | 512 | 0.691 | 0.550 | 0.328 | 1.681 | 2.003 | 2.879 |
| SSM + Additive | 2048 | 0.682 | 0.593 | 0.294 | 1.679 | 1.659 | 3.004 |

latent-space metrics at the longest horizon, suggesting that its surprise-based memory update captures something useful about long-range structure that fixed-state and cache-based encoders do not.

## B.1 Effect of SSM State Size

To test whether the SSM encoder's performance relative to the mLSTM is a matter of capacity, we enlarge its recurrent state fourfold, from $S{=}512$ to $S{=}2048$, and widen the selective-scan $\Delta$-projection (rank $\to 512$), for both the Prepend and Additive injectors. As shown in Table 4, quadrupling the state does not improve recall: SSIM is no better at either injection type and cycle error is not reduced, rising at the 50-step horizon. The SSM remains well below the mLSTM (h=10 SSIM $\approx 0.78$) regardless of state size, so the gap is not closed by adding capacity, consistent with Proposition 1 (Appendix C.1).

## C Formal Analysis of Memory Mechanisms

### C.1 Representational Capacity of Matrix vs. Diagonal Memory

We formalize the structural difference between the SSM and mLSTM memory encoders that underlies the performance gap observed in Table 1 and Figure 2.

**SSM (diagonal state).** The Mamba encoder maintains a hidden state $\mathbf{h}_t \in \mathbb{R}^d$ that evolves via a diagonal state transition (Equation 6):

$$\mathbf{h}_t = \mathrm{diag}(\bar{\mathbf{a}}_t)\,\mathbf{h}_{t-1} + \bar{\mathbf{b}}_t \odot \mathbf{z}_t, \tag{15}$$

where $\bar{\mathbf{a}}_t, \bar{\mathbf{b}}_t \in \mathbb{R}^d$ are the discretized parameters and $\mathbf{z}_t \in \mathbb{R}^d$ is the input at time $t$. Unrolling over $T$ steps gives

$$\mathbf{h}_T = \sum_{k=1}^{T} \Big( \prod_{j=k+1}^{T} \mathrm{diag}(\bar{\mathbf{a}}_j) \Big) \bar{\mathbf{b}}_k \odot \mathbf{z}_k. \tag{16}$$

Because the product of diagonal matrices is again diagonal, each dimension $i$ of $\mathbf{h}_T$ depends only on the history of dimension $i$:

$$[\mathbf{h}_T]_i = \sum_{k=1}^{T} \Big( \prod_{j=k+1}^{T} [\bar{\mathbf{a}}_j]_i \Big) [\bar{\mathbf{b}}_k]_i\, [\mathbf{z}_k]_i. \tag{17}$$

The retrieval map $\mathbf{q} \mapsto \mathbf{C}_t \mathbf{h}_t$ (where $\mathbf{C}_t = \mathrm{diag}(\mathbf{c}_t)$ in Mamba's parameterization) is therefore also diagonal.

Define the *retrievable function class* of an encoder as the set of linear maps from a query $\mathbf{q}$ to the retrieved memory output. For the SSM encoder:

$$\mathcal{F}_{\mathrm{SSM}} = \big\{ \mathbf{q} \mapsto \mathrm{diag}(\mathbf{w})\,\mathbf{q} : \mathbf{w} \in \mathbb{R}^d \big\}, \tag{18}$$

a $d$-dimensional family of diagonal linear maps. **No cross-channel correlations are representable**: the output in dimension $i$ is independent of the query in dimension $j \neq i$.

**mLSTM (matrix state).** The mLSTM encoder maintains a matrix-valued cell state $\mathbf{C}_t \in \mathbb{R}^{d \times d}$ updated via outer products (Equation 7):

$$\mathbf{C}_t = f_t\, \mathbf{C}_{t-1} + i_t\, \mathbf{v}_t \mathbf{k}_t^\top. \tag{19}$$

Unrolling over $T$ steps (with $F_{k:T} := \prod_{j=k+1}^{T} f_j$):

$$\mathbf{C}_T = \sum_{k=1}^{T} F_{k:T}\, i_k\, \mathbf{v}_k \mathbf{k}_k^\top. \tag{20}$$

Each summand is a rank-1 matrix, so $\mathrm{rank}(\mathbf{C}_T) \leq T$ (before forgetting reduces it further). Memory retrieval via $\tilde{\mathbf{M}} = \mathbf{C}_T \mathbf{q}$ produces:

$$\tilde{\mathbf{M}} = \sum_{k=1}^{T} F_{k:T}\, i_k\, \langle \mathbf{k}_k, \mathbf{q} \rangle\, \mathbf{v}_k, \tag{21}$$

a weighted combination of all stored value vectors $\{\mathbf{v}_k\}$, where the weights depend on the *full* query vector through the inner product $\langle \mathbf{k}_k, \mathbf{q} \rangle$. The retrievable function class is:

$$\mathcal{F}_{\mathrm{mLSTM}} = \big\{ \mathbf{q} \mapsto \mathbf{C}\,\mathbf{q} : \mathbf{C} \in \mathbb{R}^{d \times d},\ \mathrm{rank}(\mathbf{C}) \leq T \big\}, \tag{22}$$

a family parameterized by up to $\min(T, d) \cdot d$ independent scalar values (the entries of the rank-$\min(T, d)$ matrix), compared to $d$ for the SSM.

**Proposition 1** (Cross-channel gap). *Let $d \geq 2$ and $T \geq 2$. Then:*

    *1. $\mathcal{F}_{SSM} \subset \mathcal{F}_{mLSTM}$, i.e., every diagonal retrieval map is a special case of a matrix retrieval map.*

2. *There exist retrieval targets in $\mathcal{F}_{mLSTM} \setminus \mathcal{F}_{SSM}$ that require cross-channel correlations. Concretely, any map of the form $\mathbf{q} \mapsto \mathbf{v}\mathbf{k}^\top \mathbf{q}$ with $\mathbf{v} \nparallel \mathbf{k}$ lies in $\mathcal{F}_{mLSTM}$ but not in $\mathcal{F}_{SSM}$.*

**Remark 1.** *The retrieval operation $\mathbf{C}_T \mathbf{q}$ in mLSTM is formally analogous to a single-head linear attention over stored key-value pairs $\{(\mathbf{k}_k, \mathbf{v}_k)\}_{k=1}^T$ with exponential decay weighting. This connection has been noted in Beck et al. (2024). The proposition above makes precise that this attention-like cross-channel interaction is* provably absent *in the SSM encoder's diagonal parameterization, regardless of how the input-dependent gating is configured.*

## C.2 Recency Structure of Memory Retention

We formalize why the Titans encoder's surprise-based retention mechanism may be misaligned with the temporal structure of next-frame prediction in world models.

**Optimal retention prior for next-frame prediction.** In a world model predicting $\mathbf{X}_{t+1}$ from historical context, the relevance of a past observation $\mathbf{X}_{t-k}$ is governed by the conditional mutual information $I(\mathbf{X}_{t+1}; \mathbf{X}_{t-k} \mid \mathbf{X}_{t-k+1:t})$. For most physical environments with smooth dynamics, this quantity decreases with $k$. Recent frames carry the most predictive information about the next frame, and the marginal contribution of older frames diminishes due to the Markov-like structure of the transition dynamics. An ideal memory encoder for next-frame prediction should therefore assign monotonically decreasing retention weights as a function of temporal distance $k$.

**mLSTM: exponential recency decay.** The mLSTM's cell state at time $T$ retains the contribution of a value $\mathbf{v}_k$ written at step $k$ with weight (Equation 20):

$$w_k^{\mathrm{mLSTM}} = i_k \prod_{j=k+1}^{T} f_j, \tag{23}$$

where $f_j = \exp(\tilde{f}_j) > 0$ are the forget gates. Since each factor satisfies $0 < f_j < \infty$ (and in practice $f_j < 1$ for most timesteps to prevent unbounded growth), the product $\prod_{j=k+1}^{T} f_j$ decays geometrically in the age $T - k$. This imposes a monotonic recency bias by construction. This means more recent items always receive higher retention weight, all else being equal. The rate of decay is input-dependent and learnable through $\tilde{f}_j$, allowing the model to adapt the effective memory span to the dynamics of the environment.

**Titans: surprise-based retention.** The Titans encoder updates its memory weights $\mathbf{W}_t$ via:

$$\mathbf{W}_t = \mathbf{W}_{t-1} - \theta_t \odot \nabla_{\mathbf{W}} \mathcal{L}(f_{\mathbf{W}_{t-1}}, \mathbf{Z}_t), \tag{24}$$

where $\theta_t$ is a learned forget gate and $\mathcal{L}$ is the surprise loss. The key observation is that the gradient $\nabla_{\mathbf{W}} \mathcal{L}$ is proportional to the prediction error of the current memory on the new input, not to the temporal distance of stored items. Consequently, high-surprise items, that induced large gradients when first encountered, leave persistent traces in $\mathbf{W}_t$ regardless of how long ago they were written. Formally, the retention of a memory item from step $k$ is governed by how much the subsequent gradient updates at steps $k + 1, \ldots, T$ overwrote the weight change induced at step $k$. This depends on the content of all subsequent inputs (through their surprise values), not on their temporal distance from step $k$.

**Implications for world modeling.** In environments where scene dynamics are approximately Markovian at the frame level, as in MemoryMaze, the mLSTM's recency bias is structurally aligned with the predictive task. Titans' content-dependent retention may preserve surprising but temporally distant observations (e.g., a distinctive wall color seen many steps ago) at the expense of the most recent and most predictive context. This misalignment may compound during imagination rollouts, where the model must autoregressively predict future frames using its own outputs, amplifying any degradation from suboptimal memory content.

