# OpenReview forum: "On Memory: A Comparison of Memory Mechanisms in World Models"
_TMLR — Under review for TMLR_

### Review · Reviewer_ftAi · 2026-05-31

**Summary Of Contributions:**

The paper proposes a taxonomy and empirical study of memory mechanisms for transformer-based world models, splitting designs into two orthogonal parts: memory encoding and memory injection, followed by a controlled comparison of 20 combinations in MemoryMaze. The authors find that an mLSTM-based memory with additive injection gives the best accuracy–compute trade-off on a long-horizon recall benchmark.

Strengths

S1) Paper is clearly written, the idea is easy to understand and the abstract conveyed the central messages.

S2) The metrics are diverse enough (LPIPS, MSE, CycleMSE etc.) to prove the claims on the superior results on mLSTM.

Weaknesses

W1) The entire empirical study is on MemoryMaze, which feels limit and makes me question if the same ranking holds in another environment. There are other toy environments on memory e.g. BlockWorld from Flow Equivariant World Models (ICML 2026) that might be suitable to be one of the environment candidates. The authors may need to justify why only MemoryMaze is selected.

W2) The taxonomy has a methodological limit and it did not include KV-cache-based methods, or retrieval-based memory, which makes the title claim "On Memory: A Comparison of Memory Mechanisms in World Models" a little bit weak. The authors may need to consider on adjusting the framing.

**Audience:**

Yes

**Audience Explanation:**

Interesting empirical study on world model memory design ablations

**Broader Impact Concerns:**

/

**Claims And Evidence:**

Yes

**Claims Explanation:**

The study is overall convincing, and with detailed analysis on the results, but the entire empirical study is on a single environment, which feels limited. However it reads as an interesting findings paper and the taxonomy is potentially useful and can be used as ablation in future studies.

**Requested Changes:**

I feel the paper can be further improved if the authors also bring up the core findings in the introduction, e.g. showing the prepend and additive approaches proved to be in the top-performing configs, also the mLSTM superior's role.

Figure 3 is a bit hard to see, it can be represented as a relative scale to common baseline, with a number on top and how much gain / loss compared to baseline, or add a red dotted line as baseline like Figure 6.

Also W1 and W2

---

> ### Author Response · Authors · 2026-07-06
> **Response to Reviewer ftAi's comments**
>
> Thank you for your helpful suggestions and for taking the time to review our paper. Below are some clarifications and addressed comments from your feedback:
>
> *Single environment MemoryMaze*
> - Please also see the similar responses to the other reviewers regarding MemoryMaze. We acknowledge this in the limitations section but have strengthened the language to make it clear. MemoryMaze is a standard memory-recall benchmark used by the works we build upon, and its randomized wall colors and object placements are designed specifically to stress recall. We deliberately scoped the study to a single controlled environment so that performance differences are attributed to the memory mechanism rather than to environment-specific factors. We agree that evaluating additional environments such as BlockWorld is a valuable next step, and we will cite it as a concrete target for future work.
>
> *KV-cache clarification*
> - We'd like to clarify that KV-cache-style memory is included: our Cache encoder is a sliding buffer of raw latents (Eq. 3, following Transformer-XL and WorldMem), and the Cache+Prepend combination is exactly the attend-over-cache mechanism. What we do not vary is the selection/retrieval policy over that buffer, which is an orthogonal third axis. We updated the text to make the KV-cache instantiation explicit in Section 4.1, added the selection axis to the discussion, and added a scoping sentence so the framing reads as a comparison of memory compression and injection mechanisms rather than retrieval policies.
>
> *Core findings in introduction*
> - As you suggested, we've included a new paragraph in the introduction to bring forward the core findings to help the reader.
>
> *Updates for Figures 2-3*
> - We agree that a relative comparison to baseline is a much better format to portray the results in Figures 2-3, so we've replaced them with the updated figures and added the numbers above the bars.

---

### Review · Reviewer_CerT · 2026-06-10

**Summary Of Contributions:**

The paper presents an extensive comparative study of memory mechanisms in transformer-based world models, with the goal of enabling longer imagined rollouts without drift or hallucination. The authors decompose a memory mechanism into two components: memory encoding and memory injection. The encoding methods are a Cache, Titans (online neural weights), a Mamba SSM hidden state, and a matrix-LSTM (mLSTM) cell; the injection methods are Prepend, Additive, Cross-Attention, AdaNorm, and LoRA/QK-modulation. The authors then evaluate the resulting combinations on the MemoryMaze two-room environment over imagination horizons of 10, 20, and 50 steps. The main conclusion is that the matrix-LSTM (mLSTM) encoder, paired with the lightweight Additive injection, gives the best overall quality.

**Audience:**

Yes

**Audience Explanation:**

Memory mechanisms for world models, and the broader comparison between transformer- and state-space-based memory, are topics of active interest to the TMLR audience. The encoding/injection taxonomy and the practical recommendation are useful to researchers and practitioners working on world models and long-context sequence models.

**Broader Impact Concerns:**

There is no significant ethical concerns with this work.

**Claims And Evidence:**

Yes

**Claims Explanation:**

The claims are backed by an extensive and well-organized comparative study that covers all encoder–injection combinations across multiple imagination horizons and several reconstruction and latent-fidelity metrics.

**Requested Changes:**

1. Account for memory size in the comparison. The study does not control for the size of each memory representation. The mLSTM's matrix-valued cell state is substantially larger than the SSM or Titans states, so its advantage may come from greater memory capacity rather than from its matrix structure. The authors should report the memory size of each encoder and, ideally, compare the encoders under a matched memory budget.

2. Discuss the connection to memory in continual learning. The proposed encoding/injection framing closely mirrors the well-established notions of memory encoding and replay/recall in continual learning. The paper would be stronger if it discussed this relationship and the related literature [1, 2].

3. Discuss memory selection. Memory selection — deciding which past samples to store and replay — is a central and effective idea in continual learning [1, 2, 3], even though it has been less explored in world models. The authors should discuss how memory selection relates to their encoders and whether it could further improve memory mechanisms in world models.


[1] Van de Ven, Gido M., Hava T. Siegelmann, and Andreas S. Tolias. "Brain-inspired replay for continual learning with artificial neural networks." Nature Communications 11.1 (2020): 4069.

[2] Yin, Peng, et al. "BioSLAM: A bioinspired lifelong memory system for general place recognition." IEEE Transactions on Robotics 39.6 (2023): 4855–4874.

[3] Smith, James Seale, et al. "Adaptive memory replay for continual learning." Proceedings of the IEEE/CVF Conference on Computer Vision and Pattern Recognition (CVPR), 2024.

---

> ### Author Response · Authors · 2026-07-06
> **Response to Reviewer CerT's Comments**
>
> Thank you for your suggestions to improve the paper. We'd like to address some of your feedback below:
>
> *Accounting for memory size*
> - The three encoders (mLSTM, SSM, and Titans) all compress the past into a fixed-size state, whereas the Cache memory retains only the last K raw latents. These encoders differ in the type of memory they maintain rather than only in size. For example, the SSM keeps a diagonal vector state, the mLSTM a full matrix cell, Titans a set of online-updated MLP weights, and the Cache a raw buffer of the last K latents. The mLSTM's matrix state is intrinsically larger than the SSM's vector, but that is a consequence of using a matrix rather than a diagonal memory. Matching scalar budgets across these types is therefore not straightforward, requiring the SSM's state with the mLSTM's capacity would require an unnaturally large state. We test this empirically (in Appendix B.1) by increasing the SSM’s state size from 512 to 2048 along with a wider selective scan projection of 512. We find that the larger SSM state performs slightly worse than the original, suggesting a much larger state size would show diminishing returns, given we had the compute capacity to support this experiment in the review window.
>
> *Connection to continual learning*
> - We agree, and have added a related-work subsection connecting our encoding/injection axes to the encoding and replay/recall distinction in continual learning, with the corresponding citations.
>
> *Memory selection/retrieval*
> - This is a good point and connects to the question about retrieval-based memory raised by Reviewer ftAi. Memory retrieval/selection is an orthogonal to the encoding/injection components we address here in this study. Given methods of encoding and injecting the memory, the adjacent component is then selecting memory from history. In our study, we hold the selection fixed so that every encoder sees the full window, which isolates the encoding and injection effects. We will, however, add a paragraph positioning selection as a third axis but clearly state that this analysis is focused on analyzing encoding and injection in isolation.

---

### Review · Reviewer_A58N · 2026-06-28

**Summary Of Contributions:**

This paper presents a systematic empirical study of memory augmentation mechanisms for transformer-based world models. The authors propose a taxonomy decomposing memory into two orthogonal axes: encoding (how past information is compressed) and injection (how compressed memory is reintroduced into the residual stream), and evaluate all 20 combinations of 4 encoders × 5 injectors on the MemoryMaze benchmark across imagination horizons of 10, 20, and 50 steps. The central finding is that mLSTM encoding paired with Additive injection achieves the best overall tradeoff between reconstruction quality, latent consistency, and computational cost, matching or exceeding a pure Mamba backbone.

**Audience:**

Yes

**Audience Explanation:**

- The paper introduces a clean decomposition of memory systems into encoding and injection components. This abstraction unifies a wide variety of existing memory approaches and provides a useful conceptual framework for future work.

**Claims And Evidence:**

Yes

**Claims Explanation:**

- The evaluation protocol explicitly measures memory recall over imagination horizons of varying length and includes both image-space and latent-space metrics.
- A surprising and practically important result is that the compressed mLSTM memory outperforms both cache-based retrieval and Mamba-style state-space memories. This challenges the common intuition that preserving raw historical context should always be superior and provides useful evidence for learned memory compression.

**Requested Changes:**

- The experiments are conducted using a relatively small four-layer ViT with a short context window. It remains unclear whether the observed rankings between memory mechanisms would persist at the scales used in contemporary world models, where larger transformers and longer contexts may alter the tradeoffs substantially.
- Several key conclusions such as mLSTM outperforming SSMs due to cross-channel interactions or explicit forgetting gates are presented as hypotheses rather than supported by experiments.
- The paper evaluates memory recall quality but does not test whether improved memory leads to better planning, reinforcement learning performance, or model predictive control.
- All experiments are conducted in a single environment (MemoryMaze two-room) and on a single task (recent state recall). It is therefore unclear whether the conclusions generalize to more realistic world modeling settings involving complex dynamics, larger scenes, embodied control, or video generation.
- This is probably not a weakness point but I would like to flag that Figure 1 appears stylistically similar to claude-generated figure. While there's no explicit rules from TMLR to prohibit authors from using LLMs or generative AI to create figures, this figure could be improved to further communicate the paper's key technical ideas effectively.

---

> ### Author Response · Authors · 2026-07-06
> **Response to Reviewer A58N's comments**
>
> Thank you for your helpful suggestions for improving our paper. Our responses to your comments are as follows:
>
> *Small 4-layer ViT*
> - We agree that studying these tradeoffs at larger scale is valuable and belongs in future work. The 4-layer backbone for the model, however, is sufficient for MemoryMaze’s 64x64 images. This study is deliberately scoped as a controlled comparison, where performance differences are attributed to the memory mechanism rather than to backbone model scale. This also supports using a smaller backbone plus memory mechanism in place of a larger backbone when compute is constrained. We have revised the Limitations and Conclusion to state this scope explicitly.
>
> *Key conclusions read as hypotheses*
> - We have revised the relevant text to present these explanations as hypotheses grounded in our results and flagged as topics for dedicated future study. We note that the underlying capabilities are investigated formally in the Appendix, while the in depth study of each property merits its own full-length study.
>
> *No downstream planning, RL, MPC*
> - We’d like to emphasize that we intentionally decided to scope the study to control for memory’s added benefit alone without the complexity of planning or policy performance. We did this to clearly isolate the effects of each memory mechanism on recall. The downstream performance is a clear next step that we encourage future work to focus on.
>
> *Figure 1 appears LLM-generated*
> - Figure 1 is intended to visually show where along the transformer’s residual stream the memory injection and encoding mechanisms lies to show the reader the various forms of integrating memory into a transformer. While we acknowledge that AI-tools were used to assist in the design of Figure 1, all other figures and text were created manually, and the use of AI for the Figure 1 is due to 1) the lack of a sufficient subscription for high quality figure editing tools and 2) my complete lack of artistic ability.

---

### Author Response · Authors · 2026-07-06
**Response to all Reviewers**

We’d like to thank all reviewers for their reviews and suggestions to improve the paper. We have uploaded a new manuscript with revisions highlighted in red and updated figures 2-3 with relative performance framing, as suggested by Reviewer ftAi.


The main changes are summarized below, with point-by-point responses following.

- Resurfaced the core findings in the introduction to provide the reader with a clear summary of the results early on. We added a new contributions paragraph that states that mLSTM is the strongest encoder across metrics and horizons and that Prepend and Additive are the best injectors.
- Added discussion of the link to continual-learning and how retrieval-based memory relates to our encoding/injection framework as a third, orthogonal axes of memory design.
- Updated Figures 2-3 framing to be relative to the baseline reference line with annotations for each bar for readability.

**Motivation for MemoryMaze environment**:
*  MemoryMaze is a standard memory-recall benchmark used by the works we build upon, and its randomized wall colors and object placements are designed specifically to stress recall. We now make our motivation for using it explicit in the text and explain why it suffices for isolating the contribution of memory mechanisms to recall.

**Downstream task performance**:
* We’d like to emphasize that we intentionally decided to scope the study to control for memory’s added benefit alone without the complexity of planning or policy performance. We did this to clearly isolate the effects of each memory mechanism on recall. The downstream performance is a clear next step that we encourage future work to focus on.